# Signalling through AMPA receptors on oligodendrocyte precursors promotes myelination by enhancing oligodendrocyte survival

**Eleni Kougioumtzidou[1], Takahiro Shimizu[1†], Nicola B Hamilton[2†], Koujiro Tohyama[3†], Rolf Sprengel[4], Hannah Monyer[5], David Attwell[2\*], William D Richardson[1\*]**

[1]Wolfson Institute for Biomedical Research, University College London, London, United Kingdom; [2]Department of Neuroscience, Physiology and Pharmacology, University College London, London, United Kingdom; [3]The Center for Electron Microscopy and Bio-Imaging Research and Department of Physiology, Iwate Medical University, Morioka, Japan; [4]Max-Planck Research Group at the Institute for Anatomy and Cell Biology, University of Heidelberg, Heidelberg, Germany; [5]Department of Clinical Neurobiology, Deutches Krebforschungzentrum, University of Heidelberg, Heidelberg, Germany

**\*For correspondence:** d.attwell@ucl.ac.uk (DA); w.richardson@ucl.ac.uk (WDR)

[†]These authors contributed equally to this work

**Competing interests:** The authors declare that no competing interests exist.

**Abstract** Myelin, made by oligodendrocytes, is essential for rapid information transfer in the central nervous system. Oligodendrocyte precursors (OPs) receive glutamatergic synaptic input from axons but how this affects their development is unclear. Murine OPs in white matter express AMPA receptor (AMPAR) subunits GluA2, GluA3 and GluA4. We generated mice in which OPs lack both GluA2 and GluA3, or all three subunits GluA2/3/4, which respectively reduced or abolished AMPAR-mediated input to OPs. In both double- and triple-knockouts OP proliferation and number were unchanged but ~25% fewer oligodendrocytes survived in the subcortical white matter during development. In triple knockouts, this shortfall persisted into adulthood. The oligodendrocyte deficit resulted in ~20% fewer myelin sheaths but the average length, number and thickness of myelin internodes made by individual oligodendrocytes appeared normal. Thus, AMPAR-mediated signalling from active axons stimulates myelin production in developing white matter by enhancing oligodendrocyte survival, without influencing myelin synthesis per se.

## Introduction

Myelination of axons by oligodendrocytes (OLs), which develop postnatally from proliferating oligodendrocyte precursors (OPs), enables fast and energy-efficient propagation of electrical signals in the central nervous system (CNS). By decreasing the capacitance of axons, myelin dramatically increases the speed of action potentials and reduces the energy demands of the axons. Moreover, the ensheathing OLs can transfer energy substrates into axons (*Fünfschilling et al., 2012*; *Lee et al., 2012b*; *Saab et al., 2016*) possibly supporting them during periods of high metabolic demand. Synthesis of myelin requires a large investment of energy (*Harris and Attwell, 2012*), so it seems likely that myelination evolved to focus on active axons, rather than axons that are electrically silent because, for example, they are still elongating, or they are excess to requirements and destined for developmental pruning. This would require that developing OLs can sense the electrical activity of axons.

OLs that differentiate in culture from purified OPs can myelinate fixed axons and synthetic nanofibres in vitro, so electrically active axons are not essential for myelination (*Rosenberg et al., 2008*; *Lee et al., 2012a*; *Bechler et al., 2015*). Nevertheless, many in vitro and in vivo studies suggest that electrical activity can modulate myelination by influencing the proliferation and/or differentiation of OPs, thus determining the number of OLs that are available to myelinate (*Barres and Raff, 1993*; *Demerens et al., 1996*; *Li et al., 2010*; *Gibson et al., 2014*; *Fannon et al., 2015*; *Etxeberria et al., 2016*). In addition, electrical activity can influence the process of myelination by OLs (*Hines et al., 2015*; *Mensch et al., 2015*; *Wake et al., 2011*; *Lundgaard et al., 2013*; *Wake et al., 2015*; *Etxeberria et al., 2016*). OL production and myelination in juvenile or adult mice are also influenced by experience, presumably via altered neuronal activity (*Simon et al., 2011*; *Liu et al., 2012*; *Makinodan et al., 2012*; *Mangin et al., 2012*; *Sampaio-Baptista et al., 2013*; *Hill et al., 2014*; *McKenzie et al., 2014*; *Xiao et al., 2016*).

How might electrical activity be signalled to OL lineage cells? A likely mechanism is that neurotransmitters released from active axons modulate OL formation or myelination. OL lineage cells can sense the release of neurotransmitter directly, because they express ligand-gated channels (reviewed by *Larson et al., 2016*), including AMPA/kainate and NMDA receptors for the excitatory transmitter glutamate. OPs receive direct AMPA receptor (AMPAR)-mediated synaptic input from unmyelinated or partially myelinated axons in all brains regions examined so far (*Bergles et al., 2000*; *Chittajallu et al., 2004*; *Lin and Bergles, 2004*; *Lin et al., 2005*; *Ge et al., 2006*; *Kukley et al., 2007*; *Ziskin et al., 2007*; *Káradóttir et al., 2008*; *Müller et al., 2009*). Synaptic input is maintained during OP division (*Kukley et al., 2008*; *Ge et al., 2009*) but is down-regulated as OPs differentiate into OLs (*Cahoy et al., 2008*; *De Biase et al., 2010*; *Kukley et al., 2010*), suggesting that it might be important in the control of OL development.

In support of the idea that neurotransmitters regulate myelination, blocking vesicular release from neurons in vitro (*Wake et al., 2011*, *2015*) or in genetically manipulated zebrafish (*Hines et al., 2015*; *Mensch et al., 2015*; *Koudelka et al., 2016*) reduced myelination without much affecting OL numbers. Conversely, blocking glutamate uptake into secretory vesicles in retinal ganglion cells in early postnatal mice increased the number of OLs that developed in the optic tract, suggesting that glutamate signalling negatively regulates OL development (*Etxeberria et al., 2016*). Glutamate-evoked NMDA receptor (NMDAR)-mediated signalling in OL lineage cells has been suggested to accelerate the myelination of active axons (*Lundgaard et al., 2013*), which might occur by NMDAR activation up-regulating glucose transport into developing OLs (*Saab et al., 2016*). The role of synaptic AMPAR signalling is more enigmatic; pharmacological block of AMPAR-mediated signalling in OPs in vitro promotes proliferation and differentiation of OPs (*Gallo et al., 1996*; *Yuan et al., 1998*; *Fannon et al., 2015*) but decreases myelination (*Fannon et al., 2015*). These data are hard to interpret, because pharmacological manipulations of AMPARs in vitro affect not only OPs but also neurons, which might release substances other than glutamate to regulate myelination. Hence, we still do not have direct in vivo evidence for a role of AMPAR signalling in regulating OL lineage development.

To identify effects on myelination of AMPAR signalling in OL lineage cells, it is necessary to manipulate AMPARs in those cells without affecting neuronal AMPARs. We have therefore used *Sox10-Cre* to inactivate the *Gria2* gene (encoding AMPAR subunit GluA2), or both *Gria2* and *Gria4*, specifically in OL lineage cells on a *Gria3* germline knockout background. Examination of these *Gria2/3* and *Gria2/3/4* compound mutant mice revealed that AMPAR signalling in OPs promotes the development of OLs and myelin sheaths by stimulating survival of newly-differentiating OLs.

## Results

### OPs in subcortical white matter express AMPAR subunits GluA2-4 but not GluA1

To determine which AMPAR subunits are expressed by OPs in vivo, we examined sections of postnatal mouse forebrain by in situ hybridization (ISH) for *Gria1-4*, which encode receptor subunits GluA1-4 (also known as GluR1-4 or GluRA-D). We focused on subcortical white matter (SCWM), which contains axons, glial cells and blood vessels but very few neuronal cell bodies. We found that *Gria2*, *Gria3* and *Gria4* are expressed by many small cells in the postnatal day 1 (P1) and P14 SCWM

(*Figure 1*). In contrast, *Gria1* was expressed by very few cells in the SCWM, which NeuN labelling showed were neurons (*Figure 1A–C* and not shown). As expected, *Gria1-4* were all expressed highly by many neurons throughout the forebrain (*Figure 1A*). Fluorescence ISH for individual *Gria* family members followed by immunolabelling for the transcription factor Olig2 confirmed that *Gria2-4* were expressed by OL lineage cells in white matter; Olig2 was expressed in $88 \pm 3\%$ of $Gria2^+$ cells (817 cells counted in four mice), $94 \pm 4\%$ of $Gria3^+$ cells (506 cells from three mice) and $90 \pm 0.3\%$ of $Gria4^+$ cells (560 cells from three mice) (*Figure 2A–C*). Double ISH for *Pdgfra* and *Gria2* followed by immunolabelling for Olig2 indicated that GluA2 is expressed by the majority of OPs (*Figure 2D*), consistent with previous electrophysiological findings (*Kukley et al., 2007*; *Ziskin et al., 2007*). Occasional large cells with strong *Gria* expression but no Olig2 co-labelling (*Figure 2B*) were shown to be white matter neurons (NeuN⁺, not shown). Together, these data indicate that GluA2-4 are the main AMPAR subunits expressed by OL lineage cells in the developing mouse SCWM.

 *Gria3* germline knockout (KO) mice have no overt neural phenotype (*Meng et al., 2003*). *Gria2* germline KOs have higher than normal mortality; the survivors are anatomically normal but display a range of behavioural abnormalities, including reduced exploratory and reproductive activity and impaired motor coordination (*Jia et al., 1996*; *Gerlai et al., 1998*; *Jia et al., 2001*; *Shimshek et al., 2006a*). *Gria2/3* double KOs were reported also to develop tremor, starting in the second postnatal week (*Meng et al., 2003*). Since tremor is a hallmark of dysmyelination this suggested to us that impaired AMPAR signalling in OL lineage cells might inhibit normal myelin development. We

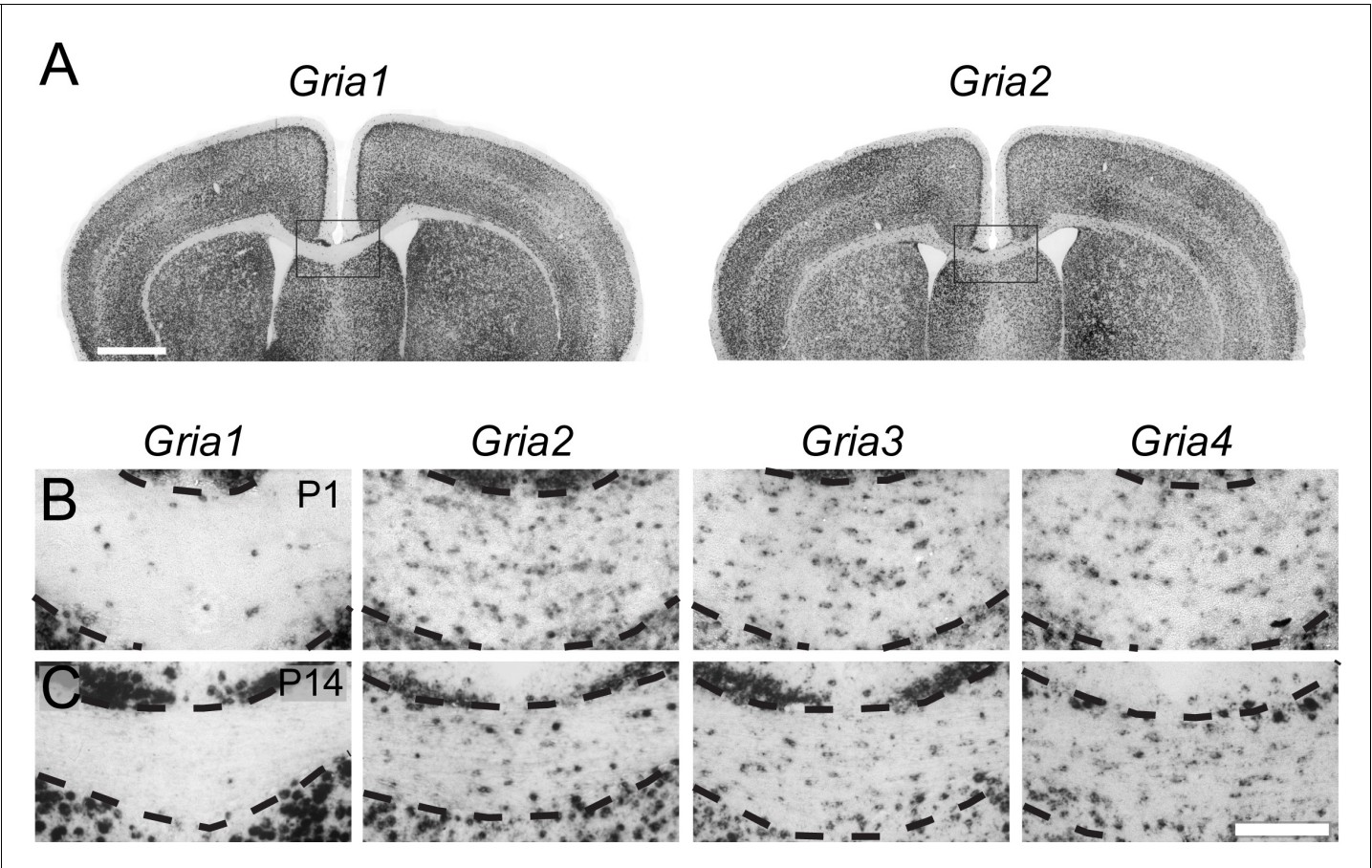

**Figure 1.** GluA2–4, but not GluA1 are expressed by presumptive OL lineage cells in the corpus callosum. (**A**) Low-magnification images of *Gria1* and *Gria2* ISH at P14. The corpus callosum (rectangles) is shown at higher magnification for *Gria1-4* (left to right) at P1 (**B**) and P14 (**C**). *Gria2*, *Gria3* and *Gria4* are expressed in many cells with small cell bodies - putative OL lineage cells - in the corpus callosum. The few *Gria1*-positive cells visible in white matter are misplaced neurons (NeuN-positive, not shown). Many forebrain neurons in grey matter strongly express *Gria1-4*. Scale bars: (**A**) 400 µm, (**B–C**) 50 µm.

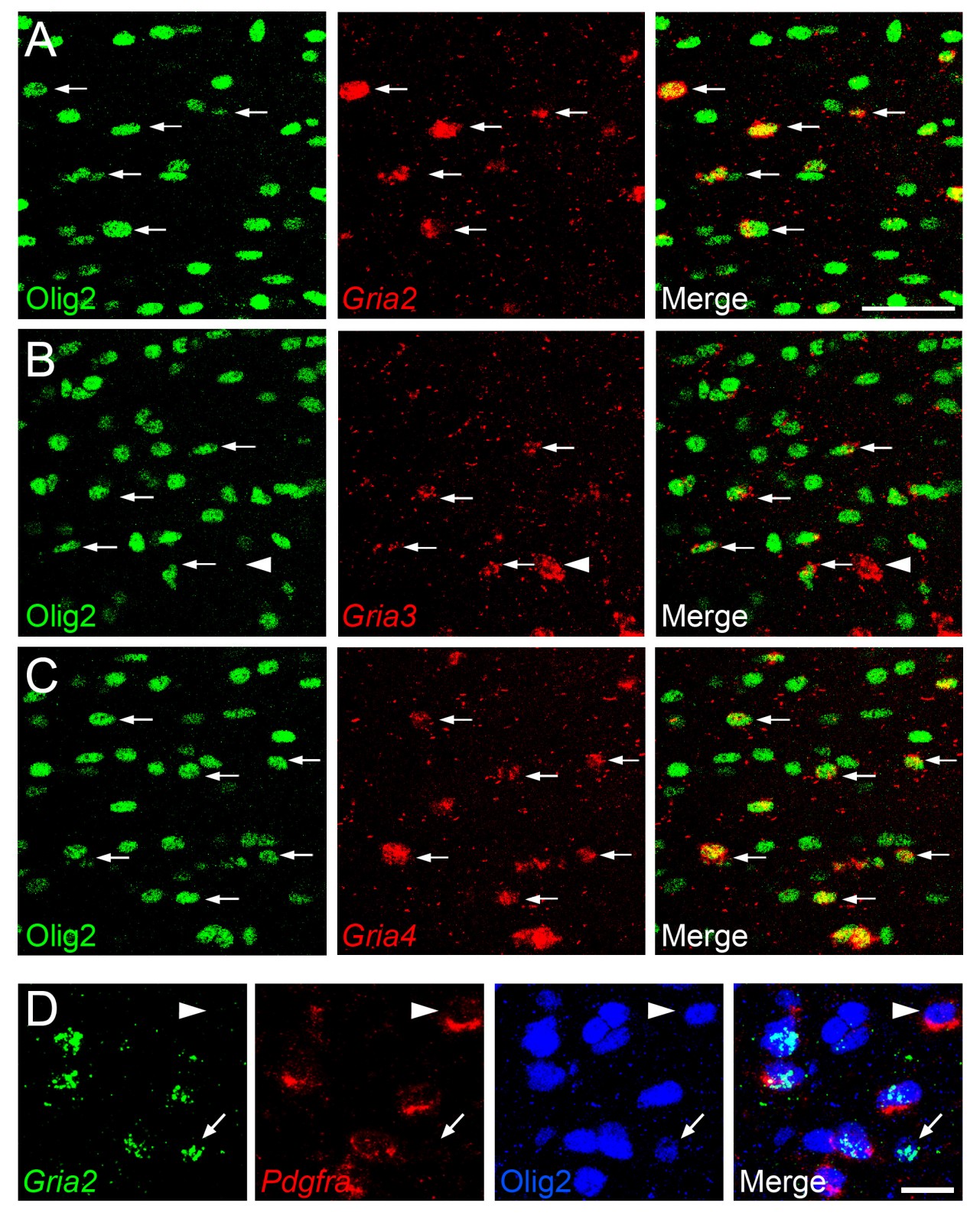

**Figure 2.** *Gria2-4* are expressed in OL lineage cells. (**A–C**) Fluorescence ISH for *Gria2-4* followed by immunohistochemistry for Olig2 in P14 corpus callosum indicates that *Gria2-4* are expressed in OL lineage cells (arrows). Occasional *Gria*+ Olig2− cells (e.g. arrowhead in middle row) are presumptive neurons. (**D**) Double fluorescence ISH for *Gria2* and *Pdgfra* followed by immunolabelling for Olig2, P9 corpus callosum. All *Gria2*+ cells are Olig2+ OL lineage cells. Many of these are (*Gria2*+, *Pdgfra*+, Olig2+) OPs. Some (*Gria2*−, *Pdgfra*+) cells (arrowhead) and (*Gria2*+, *Pdgfra*−, Olig2+) cells (arrow) are
*Figure 2 continued on next page*

*Figure 2 continued*
also present, suggesting that not all OPs express detectable levels of *Gria2* and that some differentiating OLs (*Pdgfra⁻*, Olig2⁺) continue to express *Gria2*. Scale bars: (**A–C**) 50 µm, (**D**) 25 µm.

decided to test this directly by inactivating both *Gria2* and *Gria3* in OL lineage cells. Before attempting this we checked whether deleting either subunit on its own affects OL development.

## GluA2 and GluA3 are individually dispensable for OL production

*Gria3* is X-linked in mice. To check whether *Gria3^null^* mice (i.e. *Gria3^−/Y^* males or *Gria3^−/−^* females) exhibit any obvious OL lineage-related defects, we immunolabelled P14 and P21 forebrain sections with anti-Pdgfra to detect OPs or with monoclonal CC1 (anti-adenomatous polyposis coli, APC) for differentiated OLs. The density (cells per mm²) of Pdgfra⁺ OPs in the SCWM was unaltered in *Gria3^null^* mice compared to wild type littermates at both ages (p=0.66 at P14; p=0.7 at P21, Mann-Whitney test; *Figure 3A,D*) and there was no change in the density of CC1⁺ OLs (p=0.66 at P14; p=0.7 at P21, Mann-Whitney test; *Figure 3B,E*). We examined the proliferation rate of OPs by administering 5′-ethynyl-2′-deoxyuridine (EdU) for 8 hr on P14. There was no significant difference in the fraction of Pdgfra⁺ OPs that incorporated EdU in *Gria3^null^* mice versus wild type controls (p=0.66, Mann-Whitney test; *Figure 3C,F*), indicating that the proliferation rate of OPs was not affected by loss of GluA3. We also counted the density of myelinated axons as well as their g-ratios in parasagittal sections of anterior SCWM in electron micrographs and found no difference between *Gria3^null^* mice and wild type controls at P14 (p=0.4, Mann-Whitney test; *Figure 3G,H*). Furthermore, there was no change in the frequency distribution of either myelinated or unmyelinated axon diameters (*Figure 3I,J*). Therefore, loss of GluA3 alone does not detectably affect postnatal OL production or myelination.

The GluA2 subunit is functionally distinct from other AMPAR subunits because its mRNA can be edited to change a glutamine (Q) codon to an arginine (R) codon, causing AMPAR tetramers that contain the edited 'R' version to be Ca²⁺ – impermeable. To investigate the role of GluA2 in OL development we generated conditional knockouts (cKOs) by crossing *Sox10-Cre* (*Matsuoka et al., 2005*) to *Gria2^flox/flox^* mice, in which exon 11 of *Gria2* is flanked by *loxP* sites (*Shimshek et al., 2006b*), on the *Rosa-YFP* reporter background (*Srinivas et al., 2001*). In the CNS, Sox10 expression is restricted to OL lineage cells – OPs in the embryo and both OPs and OLs postnatally (*Stolt et al., 2002*). The above cross generated *Sox10-Cre: Gria2^flox/flox^* cKOs alongside *Sox10-Cre: Gria2^flox/+^* and *Sox10-Cre: Gria2^+/+^* littermate controls (all including *Rosa-YFP*) (*Figure 4—figure supplement 1A*). We refer to these as *Gria2^−/−^*, *Gria2^+/−^* and *Gria2^+/+^*. Most OPs (~80%) in the postnatal SCWM of wild type mice originate after birth (E21/P0) from neural stem cells in the cortical ventricular zone (VZ), with the remaining ~20% of OPs migrating into the developing cortex from the ventral forebrain starting around embryonic day 16 (E16) (*Kessaris et al., 2006*; *Tripathi et al., 2011*; reviewed by *Bergles and Richardson, 2015*). Thus, *Sox10-Cre* driven recombination at the *Gria2^flox^* locus is expected to commence in the cortex and SCWM before birth and to continue after birth.

Control *Sox10-Cre: Rosa-YFP* mice (without *Gria2^flox^*) displayed widespread expression of the YFP reporter (*Figure 4—figure supplement 2*). In the SCWM at P14, 96.4% ± 1.0% of Sox10⁺ cells expressed YFP (5551 cells counted in four mice; *Figure 4—figure supplement 2A*). In the cortex, 97.1 ± 0.7% of Sox10⁺ cells expressed YFP (2506 Sox10⁺ cells counted in four mice; Supplementary *Figure 4—figure supplement 2B*), confirming that Cre was active in the great majority of OL lineage cells in both grey and white matter. YFP expression was restricted to Sox10⁺ cells in SCWM (98.1 ± 0.5% of YFP⁺ cells were Sox10⁺) and was also predominantly in Sox10⁺ cells in cortical grey matter (88.1 ± 1.6% of YFP⁺ cells were Sox10⁺; *Figure 4—figure supplement 2*). YFP expression was also observed in some astrocytes and neurons (NeuN⁺) in the ventral forebrain and in a small fraction of cortical projection neurons (1.5 ± 0.4%; 9894 NeuN⁺ neurons counted in three animals; *Figure 4—figure supplement 2B*) as well as some astrocytes (not quantified).

To confirm that GluA2 was deleted in *Gria2^−/−^* OPs, fluorescence-guided whole cell patch-clamp recordings from YFP⁺ OPs in the SCWM of P14 mice were compared to age-matched controls. AMPAR tetramers lacking GluA2 display inwardly-rectifying current responses to kainate and AMPA, due to a voltage-dependent block of the channel by polyamines at positive potentials (*Donevan and*

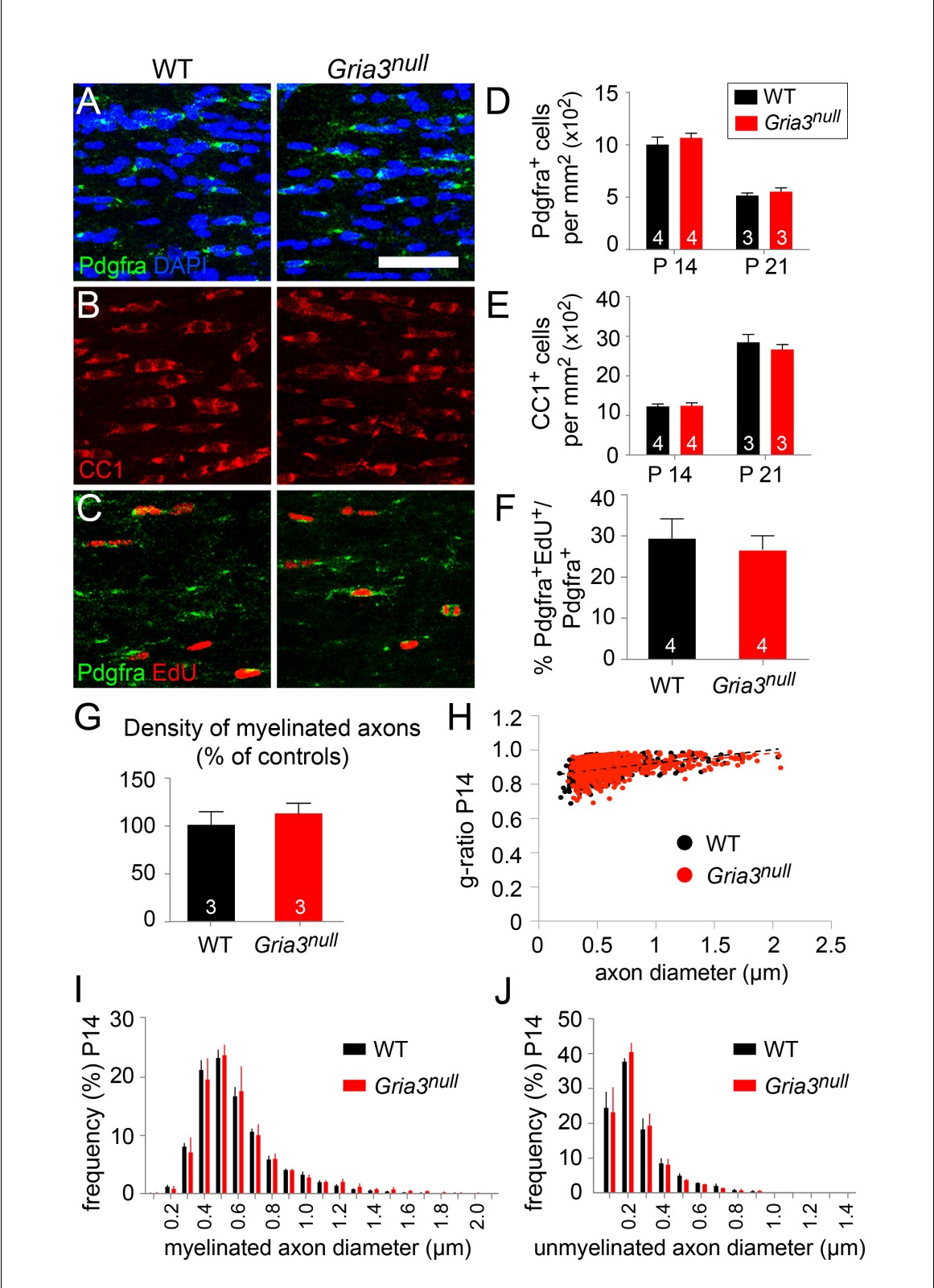

**Figure 3.** *Gria3* germline KO mice generate normal numbers of OPs and OLs. (**A, B**) Immunolabelling for Pdgfra+ OPs (**A**) or CC1+ OLs (**B**) in P14 corpus callosum of wild type (WT) and *Gria3null* mice. (**C**) EdU was administered to P14 wild type and *Gria3null* mice for 8 hr before analyzing the corpus callosum by Pdgfra immunolabelling and EdU detection. (**D, E**) Cell counts reveal no difference in the density of OPs at P14 and P21 (**D**; p=0.66 at P14, p=0.7 at P21, Mann-Whitney test; >600 cells counted in each mouse) or OLs (**E**; p=0.66 at P14, p=0.7 at P21, Mann-Whitney test; >900 cells counted in

*Figure 3 continued on next page*

*Figure 3 continued*
each mouse) in *Gria3^null* versus WT corpus callosum. (F) Cell counts reveal no difference between the EdU labelling indices of Pdgfra^+ OPs in WT versus *Gria^null* corpus callosum (p=0.66, Mann-Whitney test). (G) There was no change in the number of cross-sectional myelin figures per unit area in P14 *Gria3^null* compared to WT littermate controls (p=0.4, Mann-Whitney test). (H) Scatter plot of g-ratios as a function of axon diameter in P14 *Gria3^null* and wild type mice (>130 axons measured in each of three mice per group). (I) Frequency distribution of myelinated axon diameters from P14 *Gria3^null* and WT mice (>800 axons measured in each of three mice per group). (J) Frequency distribution of unmyelinated axon diameters from P14 *Gria3^null* and WT mice (>800 axons measured in each of three mice per group). Numbers of mice analyzed are indicated in (D), (E) and (F). Scale bar: 50 µm.

*Rogawski, 1995*). When GluA2 is present (in the edited 'R' form), the current-voltage (I-V) relationship is linear, that is, non-rectifying. Application of kainate (100 µM), which activates AMPA and kainate receptors, elicited an inward current at negative potentials in all OPs examined (*Figure 4A*). The I-V relationship of the kainate-evoked current in the presence in the patch pipette of spermine (100 µM), a polyamine that reduces outward current flow in AMPARs that lack GluA2, appeared inwardly-rectifying in all cells examined from *Gria2^{-/-}* brain slices but was almost linear in *Gria2^{+/-}* and *Gria2^{+/+}* controls (*Figure 4B,C*). The rectification index (RI), defined as (kainate-mediated current amplitude at +17 mV)/ (current amplitude at –63 mV), was significantly reduced in *Gria2^{-/-}* mice compared to *Gria2^{+/-}* and *Gria2^{+/+}* (p<10^{-4}, one-way ANOVA with Bonferroni post-hoc test; *Figure 4D*), confirming that GluA2 had been successfully deleted. Despite this, the kainate-evoked current density (current normalized to membrane capacitance, which is proportional to cell surface area) at –63 mV in *Gria2^{-/-}* OPs was not significantly reduced compared to *Gria2^{+/-}* or *Gria2^{+/+}* (p=0.73, Kruskal-Wallis test; *Figure 4E*). Thus, activation of AMPARs present at the cell surface in the presence or absence of GluA2 generates the same membrane current at –63 mV, presumably as a result of compensation from other AMPAR subunits, but the voltage-dependence of the current is different in *Gria2^{-/-}* mice (*Figure 4C*).

The density (cells per mm²) of Pdgfra^+ OPs in SCWM was unaltered in *Gria2^{-/-}* mice compared to *Gria2^{+/+}* controls at P14 (p=0.34, Mann-Whitney test; *Figure 4F,H*), as was the density of CC1^+ cells (p=0.40, Mann-Whitney test; *Figure 4F,H*). To test whether the OP proliferation rate was altered we administered EdU to P14 *Gria2^{-/-}* and *Gria2^{+/-}* mice (*Figure 4G,I*). There was no significant difference between the EdU labelling indices of Pdgfra^+ OPs in *Gria2^{-/-}* versus *Gria2^{+/-}* (p=0.94, Mann-Whitney test), indicating that the cell cycle time of OPs was not affected (*Figure 4G,I*). These results demonstrate that OL development is normal in the absence of GluA2, despite the altered AMPAR Ca^{2+} permeability.

## Double deletion of GluA2/ GluA3 in the OL lineage

To generate mice in which OL lineage cells lack both GluA2 and GluA3, we deleted *Gria2^flox* conditionally, using *Sox10-Cre*, in the *Gria3* germline KO background. We chose this approach rather than attempting a double *Gria2/Gria3* cKO because it simplifies the breeding program and it might increase the probability of Cre-mediated deletion at both *Gria2^{flox/flox}* alleles, by eliminating potential competition with *Gria3^flox*.

By appropriate breeding (*Figure 4—figure supplement 1B*) we generated mice with the genotype *Sox10-Cre: Gria3^null: Gria2^{flox/flox}: Rosa–YFP* (referred to as *Gria3^null2^{-/-}*). Control mice were *Gria3^null* with two functional copies of *Gria2*, either because they carried wild type alleles of *Gria2* (no *loxP* sites) or because they did not carry the *Sox10-Cre* transgene. In some experiments, mice with one functional copy of *Gria2* were also used as controls (*Figure 4—figure supplement 1B*). We designate these controls *Gria3^null2^{+/+}* and *Gria3^null2^{+/-}*, respectively. As observed for *Gria2^{-/-}* (see above), the kainate-evoked current in *Gria3^null2^{-/-}* was inwardly-rectifying in all YFP^+ OPs that we examined, whereas *Gria3^null2^{+/+}* and *Gria3^null2^{+/-}* controls had an almost linear I-V relationship (*Figure 5—figure supplement 1A*). The rectification index was significantly reduced in *Gria3^null2^{-/-}* mice compared to *Gria3^null2^{+/-}* and *Gria3^null2^{+/+}* controls (p<10^{-4}, Kruskal-Wallis with Dunn's post-hoc test; *Figure 5—figure supplement 1B*). These data confirm that, as for *Gria2^{-/-}* single cKOs, our genetic strategy had deleted GluA2 efficiently in OPs of *Gria3^null2^{-/-}* mice.

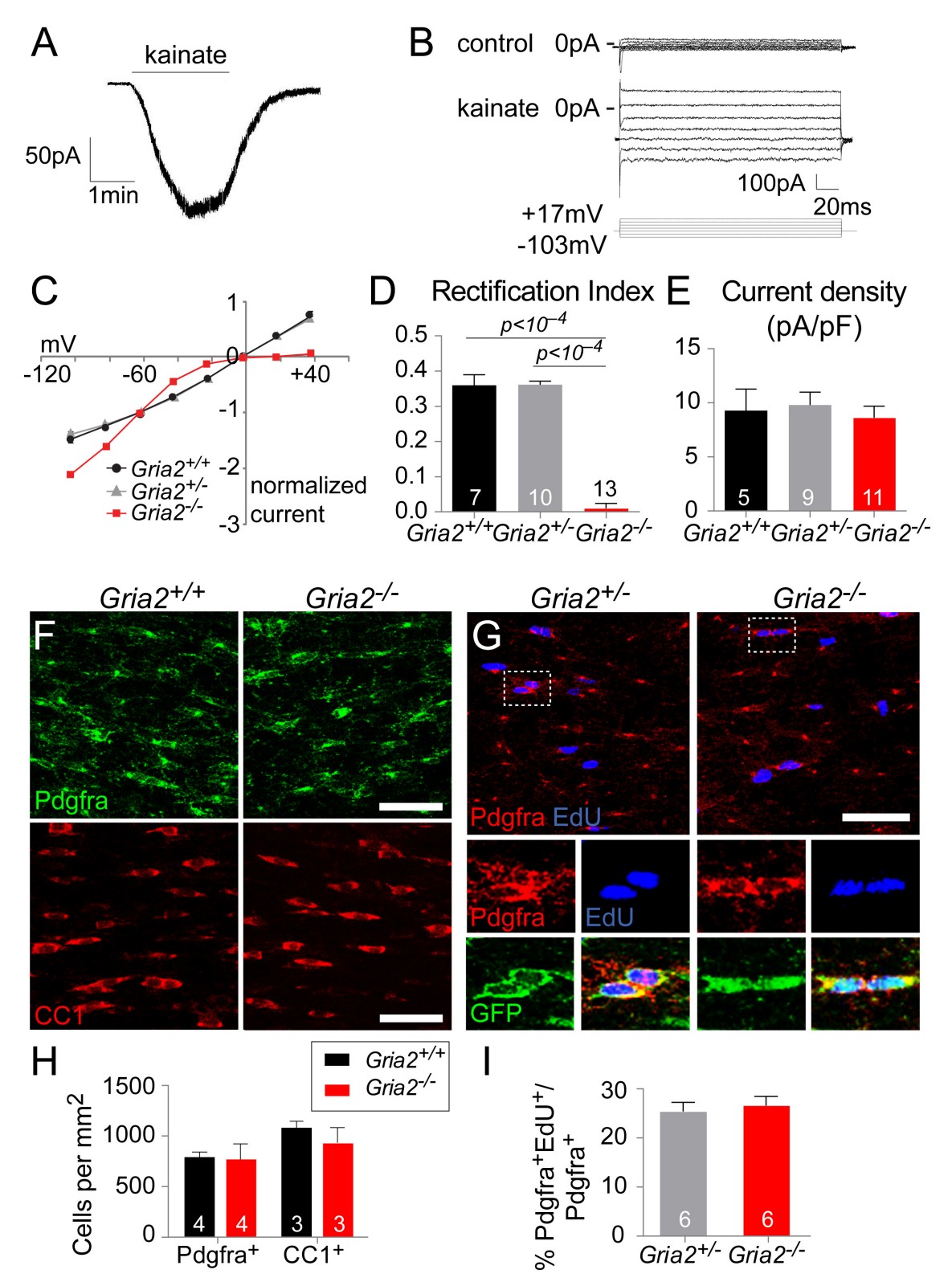

**Figure 4.** *Gria2* knock-out renders AMPARs in OPs inwardly-rectifying but does not alter numbers of OL lineage cells in white matter. (**A**) Patch-clamp recording of an OP in SCWM in an acute slice of wild type mouse forebrain, showing inward current generated by bath application of 100 μM kainate. (**B**) Currents evoked by voltage steps from –63 mV in *Gria3^null^2^+/+* OP before (control) and during application of 100 μM kainate. Voltage steps were in 20 mV increments from –103 mV to +17 mV. (**C**) Current-voltage (I–V) plots of kainate-evoked current in OPs of P14 *Gria2^-/-^* mice and

*Figure 4 continued on next page*

*Figure 4 continued*

littermate controls. With 100 µM spermine included in the patch pipette, inward rectification of the current was observed in *Gria2−/−* but not in *Gria2+/−* or *Gria2+/+*. (**D**) Rectification index (RI; current at +17 mV/ current at –63 mV) of cells from (**C**). RI is greatly reduced in *Gria2−/−* mice ($p < 10^{-4}$, one-way ANOVA with Bonferroni post-hoc test). (**E**) Mean kainate-evoked current density (current/capacitance, pA/pF) is the same in P14 *Gria2−/−*, *Gria2+/−* and *Gria2+/+* OPs (p=0.73, Kruskal-Wallis with Dunn's post-hoc test). (**F**) Immunolabelling for Pdgfra+ OPs (top) and CC1+ OLs (bottom) in P14 *Gria2−/−* and *Gria2+/+* corpus callosum. (**G**) EdU incorporation by proliferating OPs (Pdgfra+) in the P14 corpus callosum of *Gria2−/−* and *Gria2+/−* mice. The bottom panels show higher-magnification images of the cells indicated on the top; all Pdgfra+ OPs co-expressed YFP confirming that *Sox10-Cre* is expressed in all these cells. (**H**) There were no significant changes in the densities of either Pdgfra+ OPs (p=0.34, Mann-Whitney test; >600 cells counted per mouse) or CC1+oligodendrocytes (p=0.40, Mann-Whitney test; >900 cells counted per mouse) at P14. (**I**) There were no significant differences in the fractions (%) of EdU+ Pdgfra+ OPs in *Gria2−/−* versus *Gria2+/−* (p=0.89, Mann-Whitney test). Numbers of cells and mice analyzed are indicated in (**D, E**) and (**H, I**), respectively. Scale bars: 50 µm.

The following figure supplements are available for figure 4:

**Figure supplement 1.** Breeding strategies.

**Figure supplement 2.** *Sox10-Cre* drives recombination in OL lineage cells in the sub-cortical white matter and cerebral cortex.

## AMPAR-mediated currents are reduced in *Gria3null2−/−* OPs

*Gria3null2−/−* mice were viable and survived into adulthood without neurological signs, suggesting that the tremor reported in *Gria2/3* germline KO mice (**Meng et al., 2003**) was not related to impaired AMPAR signalling in OL lineage cells. Since *Gria3null2−/−* OPs lack both GluA2 and GluA3, and GluA1 appears not to be expressed in the SCWM of wild type mice (see above, first paragraph of Results; **Figure 1**), this implies that *Gria3null2−/−* OPs express GluA4 alone. Thus, it might be expected that fewer AMPARs would be assembled in OPs from *Gria3null2−/−* mice relative to controls.

To test this, we whole-cell patch-clamped YFP+ OPs in the SCWM of P14 *Gria3null2−/−* and *Gria3null2+/+* mice and measured kainate-evoked currents at –63 mV. In both genotypes, bath application of 100 µM kainate elicited inward currents, indicating that AMPA/kainate receptors were present (**Figure 5A**). Normalizing the amplitude of the maximum kainate-evoked current at –63 mV to the cell size (assessed as membrane capacitance) revealed a significant (~47%) reduction in the current density for *Gria3null2−/−* compared to *Gria3null2+/+* controls ($p < 10^{-4}$, Student's t-test; **Figure 5B**), indicating that the number of AMPARs expressed on the cell surface was decreased in *Gria3null2−/−* mice. The alternative explanation, that the single-channel current passed by the remaining AMPARs was reduced, is less likely because GluA4 homo-tetramers are reported to have a higher conductance than GluA2/GluA4 hetero-dimers, for example (**Swanson et al., 1997**).

Application of kainate activates both AMPA and kainate receptors. When a fast application system is used it is possible to distinguish the activation of each receptor type based on the kinetics of the response (**Patneau et al., 1994**); kainate receptors desensitize quickly to kainate while AMPARs desensitize slowly or not at all. With the slow application method that we employed, most of the kainate-evoked current should therefore be AMPAR-mediated. To confirm this, we examined the responses to kainate in the presence of 50 µM GYKI-52466 (GYKI), a selective AMPAR antagonist. In both *Gria3null2−/−* and *Gria3null2+/+* OPs, kainate-evoked currents were blocked by ~80% in the presence of GYKI (**Figure 5A,C**), confirming that most of the kainate-evoked current is AMPAR-mediated under our experimental conditions. Thus, in *Gria3null2−/−* mice, activation of AMPARs in OPs generates less membrane current (and hence less depolarization) than in *Gria3null2+/+* controls.

The fact that a significant fraction (~53%) of the normal kainate current is still observed in the *Gria3null2−/−* OPs (**Figure 5B**) indicates that AMPARs are still formed and trafficked to the surface membrane. It is possible either that the remaining GluA4 subunits form functional homo-tetrameric receptors, as they can do when expressed on their own in cultured mammalian cells (**Keinänen et al., 1990**), or that GluA1 is up-regulated in response to loss of GluA2 and GluA3 and forms GluA1/GluA4 heteromeric or GluA1 homomeric channels. However, by ISH we found no evidence for up-regulation of *Gria1* mRNA in *Gria3null2−/−* SCWM relative to *Gria3null2+/+* (**Figure 5— figure supplement 2**). Thus, the majority of AMPAR formed in *Gria3null2−/−* OPs are probably homo-tetramers of GluA4.

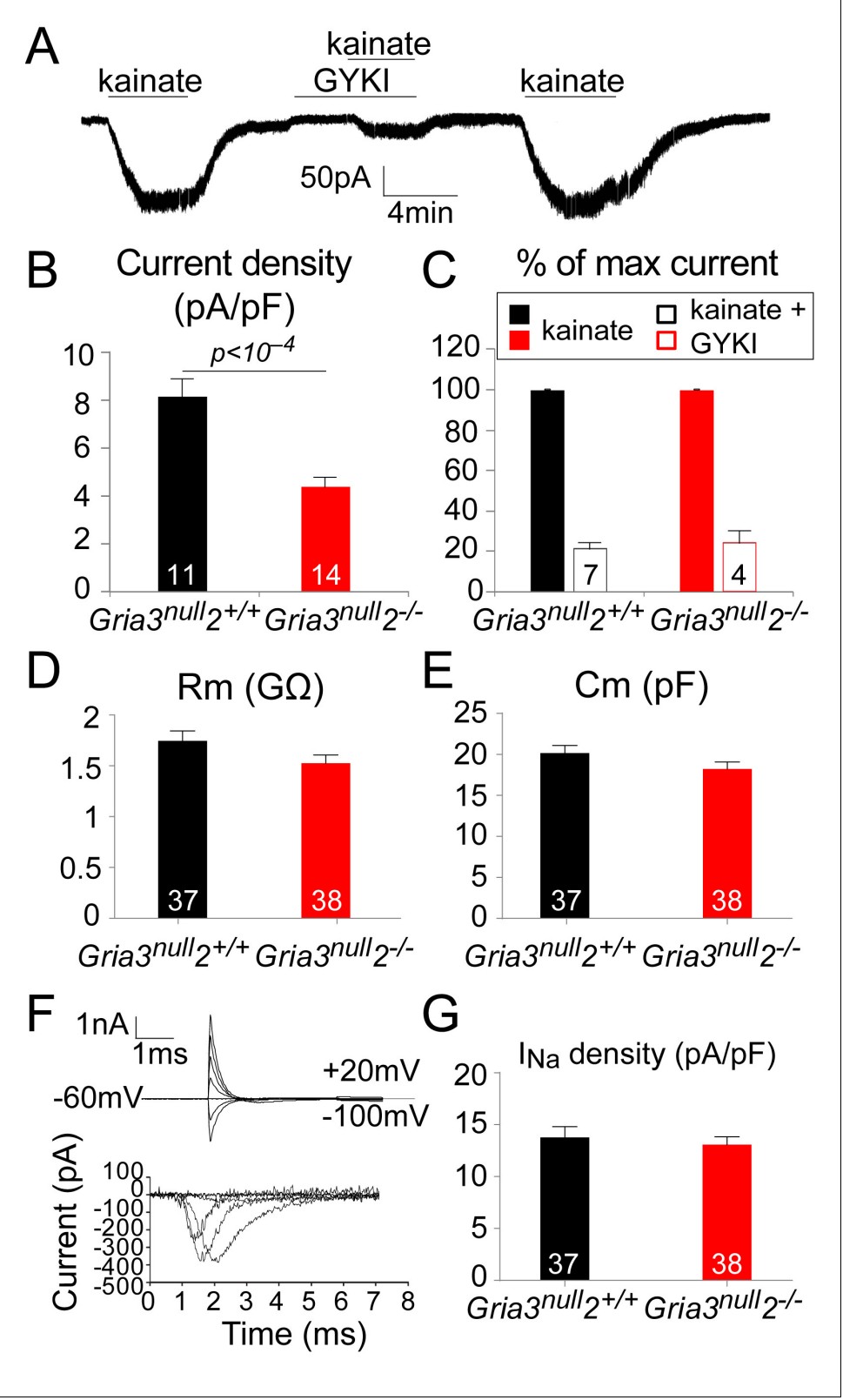

**Figure 5.** The kainate- evoked current in OPs is reduced in *Gria3null2−/−* mice. (**A**) The kainate-evoked current is blocked reversibly by the AMPAR antagonist GYKI-52466 (GYKI, 50 μM). (**B**) Mean kainate-evoked current density is ~47% less in *Gria3null2−/−* OPs compared to *Gria3null2+/+* (p<10−4, Student's t-test). (**C**) GYKI blocks the kainate-evoked current in both *Gria3null2−/−* and *Gria3null2+/+* mice (normalized to 100% before GYKI application). (**D, E**) No
*Figure 5 continued on next page*

*Figure 5 continued*

significant changes in membrane resistance Rm (D), p=0.16, Mann-Whitney test) or capacitance Cm (E), p=0.13, Student's t-test) of OPs from *Gria3^null^2^–/–* compared to *Gria3^null^2^+/+*. (F) $I_{Na}$ recorded from P14 *Gria3^null^2^+/+* OPs. Top current traces are responses to voltage steps in 20 mV increments from −63 mV, showing the capacity current and the subsequent (much smaller) ionic current changes evoked by the voltage steps. Bottom trace is after subtraction of the linearly scaled response to a 20 mV hyperpolarizing step. (G) There was no significant change in $I_{Na}$ density in *Gria3^null^2^–/–* compared to *Gria3^null^2^+/+* controls (p=0.72, Mann-Whitney test). Numbers of cells analyzed are indicated in (B–E, G).

The following figure supplements are available for figure 5:

**Figure supplement 1.** *Gria2* is successfully knocked out in *Gria3^null^2^–/–* mice.

**Figure supplement 2.** *Gria1* is not up-regulated in cells in the corpus callosum of *Gria3^null^2^–/–* mice.

---

To determine whether deletion of GluA2/3 from OL lineage cells altered their general membrane properties, the electrophysiological properties of YFP$^+$ OPs from SCWM of P14 *Gria3^null^2^–/–* and *Gria3^null^2^+/+* mice were compared (**Figure 5D–G**). Neither the membrane resistance ($R_m$) nor the capacitance ($C_m$) of *Gria3^null^2^–/–* OPs at –63 mV were different from *Gria3^null^2^+/+* controls (for $R_m$, p=0.16, Mann-Whitney test and for $C_m$, p=0.13, Student's t-test; **Figure 5D,E**). In addition, when 20 mV voltage steps were applied from a holding potential of –63 mV, membrane depolarization evoked a transient inward Na$^+$ current ($I_{Na}$; **Figure 5F**). Subtraction of the linearly scaled capacitive transient and ohmic leak current evoked by a 20 mV hyperpolarizing step allowed us to estimate the amplitude of $I_{Na}$ (**Figure 5F**). There was no significant difference in the $I_{Na}$ density between *Gria3^null^2^–/–* and *Gria3^null^2^+/+* controls (p=0.72, Mann-Whitney test; **Figure 5G**), suggesting that altering AMPAR signalling does not regulate the expression of voltage-gated sodium channels. We conclude that AMPAR signalling does not much affect the membrane properties of OPs.

## Fewer functional axon-OP synapses in Gria3^null^2^–/– white matter

The reduced kainate-evoked current in *Gria3^null^2^–/–* OPs should be mirrored by a diminished post-synaptic response at individual axon-OP synapses. To test this, we recorded evoked excitatory post-synaptic currents (EPSCs) from OPs by applying 'minimal stimulation', designed to stimulate a single axon (see Methods), to the corpus callosum. Surprisingly, the amplitude of the evoked EPSCs appeared the same in *Gria3^null^2^–/–* and their *Gria3^null^2^+/–* controls (p=0.37, Mann-Whitney test; **Figure 6A,B**). The evoked responses were blocked by GYKI (50 µM), confirming that they were AMPAR-mediated (90.3 ± 4.0% inhibition, n = 3 for *Gria3^null^2^+/–* and 89.5 ± 2.1% inhibition, n = 5 for *Gria3^null^2^–/–*).

Spontaneous synaptic currents occur at very low frequency at neuron-OP synapses but their frequency can be increased by application of the secretagogue Ruthenium Red (RR) (**Lin and Bergles, 2004**; **Kukley et al., 2007**). RR induces vesicular release at the synapse, increasing the frequency of miniature EPSCs without affecting their amplitude (**Sciancalepore et al., 1998**). Bath application of 100 µM RR caused a marked increase in the frequencies of spontaneous EPSCs recorded from OPs of *Gria3^null^2^+/+*, *Gria3^null^2^+/–* and *Gria3^null^2^–/–* mice. Consistent with the minimal stimulation-evoked EPSC data, the mean amplitude of individual RR-evoked EPSCs was not reduced in *Gria3^null^2^–/–* compared to *Gria3^null^2^+/–* controls or *Gria3^null^2^+/+* controls (p=0.32, one-way ANOVA; **Figure 6C,D**) – suggesting that, at a given synapse, post-synaptic AMPARs generate a similar current in *Gria3^null^2^–/–* as in controls. However, the frequency of RR-evoked EPSCs was ~70% less in *Gria3^null^2^–/–* than in *Gria3^null^2^+/+* controls, and was reduced to an intermediate frequency in *Gria3^null^2^+/–* controls (p=$2 \times 10^{-4}$, Kruskal-Wallis with Dunn's post-hoc test; **Figure 6E**). Consistent with previous reports (**Ziskin et al., 2007**), EPSCs were strongly inhibited by the application of 50 µM GYKI (by 88.8 ± 6.5%, n = 3 for *Gria3^null^2^–/–*, 83.3 ± 3.8%, n = 2 for *Gria3^null^2^+/–* and 93.8 ± 2.4%, n = 2 for *Gria3^null^2^+/+*), proving that they are AMPAR-mediated. The reduction in frequency of EPSCs suggests either that the probability of RR-evoked presynaptic vesicle release is decreased in *Gria3^null^2^–/–* or, more likely, that fewer synapses are made between neurons and OPs in *Gria3^null^2^–/–* mice.

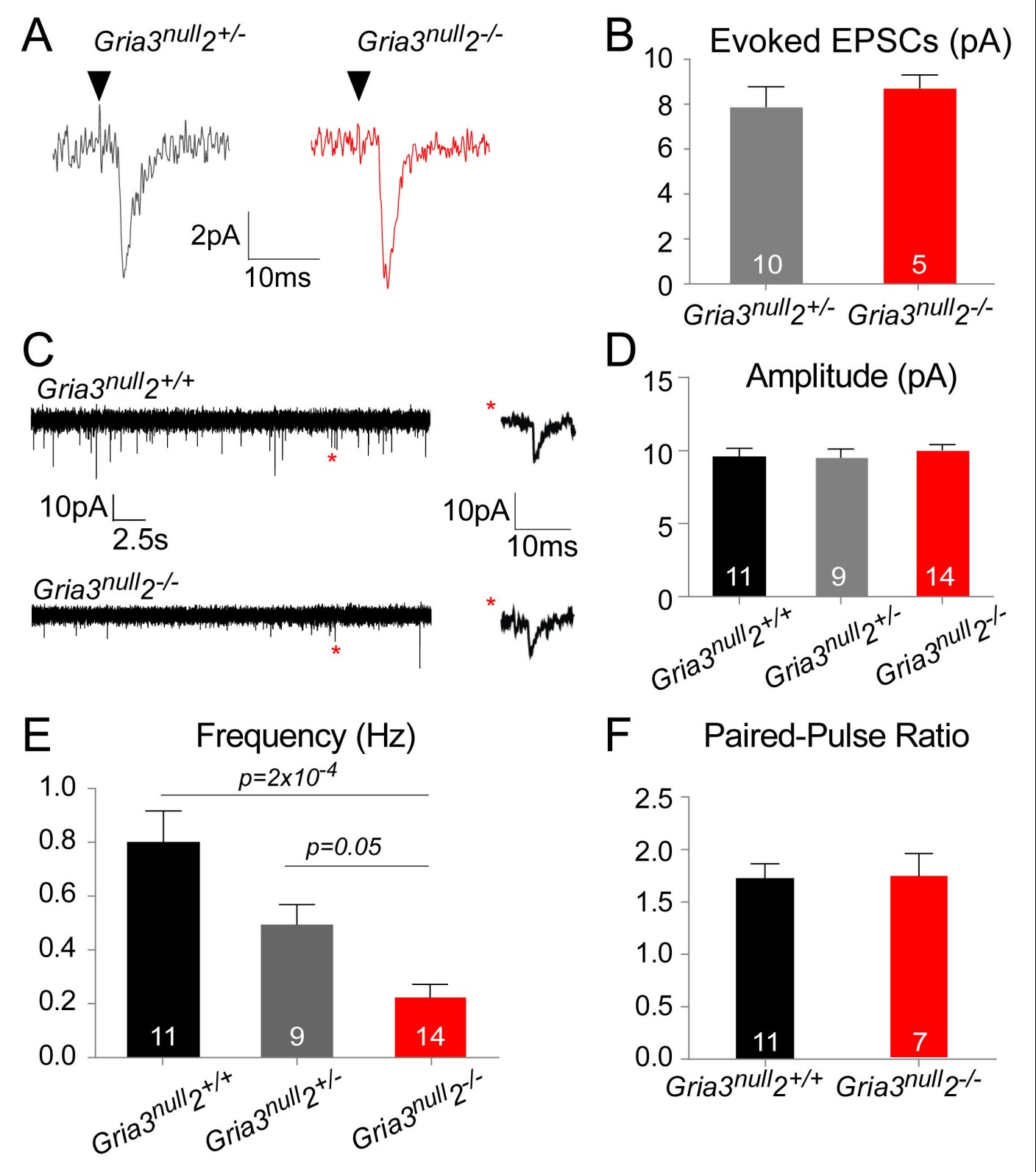

**Figure 6.** AMPAR-mediated synaptic input is reduced in *Gria3^null^2^–/–* mice. (**A**) EPSCs evoked by minimal stimulation in OPs in the SCWM of P14 *Gria3^null^2^+/–* and *Gria3^null^2^–/–* mice. Black triangle marks the time point of stimulation. (**B**) The amplitude of minimal stimulation-evoked EPSCs is not affected in *Gria3^null^2^–/–* compared to *Gria3^null^2^+/–* (p=0.37, Mann-Whitney test). (**C**) EPSCs evoked in OPs by Ruthenium Red (RR, 100 μM) in the corpus callosum of P14 *Gria3^null^2^+/+* and *Gria3^null^2^–/–* mice. Single events marked by asterisks (red) are enlarged on the right. (**D**) The amplitude of RR-evoked

*Figure 6 continued on next page*

**Figure 6 continued**

EPSCs is not affected in $Gria3^{null}2^{-/-}$ compared to $Gria3^{null}2^{+/+}$ or $Gria3^{null}2^{+/-}$ (p=0.73, one-way ANOVA with Bonferroni post-hoc test). (**E**) The frequency of RR-evoked EPSCs is ~70% less in $Gria3^{null}2^{-/-}$ compared to $Gria3^{null}2^{+/+}$ (p=2×10$^{-4}$, Kruskal-Wallis with Dunn's post-hoc test). (**F**) There was no change in the paired-pulse ratio in $Gria3^{null}2^{-/-}$ compared to $Gria3^{null}2^{+/+}$ (p=0.95, Student's t-test). Numbers of cells analyzed are indicated in (**B, D–F**).

To distinguish between these possibilities we estimated the paired pulse ratio (PPR) at neuronal-OP synapses in SCWM. The PPR was calculated from two consecutive evoked EPSCs 25 ms apart, recorded from patch-clamped OPs. There was no difference in PPR of $Gria3^{null}2^{-/-}$ OPs versus $Gria3^{null}2^{+/+}$ OPs (p=0.95, Student's t-test; *Figure 6F*). This suggests that the reduction in EPSC frequency is not due to a change in presynaptic glutamate release but rather because fewer active synapses are formed or maintained in $Gria3^{null}2^{-/-}$ mice.

## OL survival is reduced, and OL accumulation delayed, in $Gria3^{null}2^{-/-}$ white matter

Having established that OPs in $Gria3^{null}2^{-/-}$ mice receive reduced synaptic input, we asked whether the development of OL lineage cells was affected. We counted CC1$^+$ differentiated OLs in the SCWM of early postnatal (P7, P14, P21) and adult (P70) $Gria3^{null}2^{-/-}$ and $Gria3^{null}2^{+/+}$ mice. There was a ~22% reduction in the density of OLs at P7 and a ~27% reduction at P14 in $Gria3^{null}2^{-/-}$ compared to $Gria3^{null}2^{+/+}$ controls (p=0.018 at P7; p=0.001 at P14, t–tests with Holm-Bonferroni correction; *Figure 7A,B*). By P21 (p=0.14) and P70 (p=0.48, Mann-Whitney test with Holm-Bonferroni correction) this difference was no longer significant. These data suggest that production or survival of OLs is impaired in $Gria3^{null}2^{-/-}$ mice in the early postnatal period.

Enpp6 (ectonucleotide pyrophosphatase/phosphodiesterase family member 6; an ecto-enzyme with choline-specific glycerophosphodiester phosphodiesterase activity: *Morita et al., 2016*) is a recently-identified marker of newly-differentiating OLs that is strongly down-regulated in mature OLs (*Zhang et al., 2014*; *Xiao et al., 2016*). The number of Enpp6$^+$ cells present at any given time therefore provides a measure of the rate of formation of OLs from their precursors (OPs). We visualized Enpp6$^+$ cells in the SCWM of P14 $Gria3^{null}2^{-/-}$and $Gria3^{null}2^{+/+}$ mice by ISH. This revealed a ~26% reduction of Enpp6$^+$ newly-differentiated OLs in $Gria3^{null}2^{-/-}$ mice compared to their $Gria3^{null}2^{+/+}$ littermates (p=0.004, Mann-Whitney test; *Figure 7C,D*), reinforcing our conclusion that the rate of generation or survival of newly-forming OLs is reduced by loss of GluA2/GluA3. That is, glutamate signalling through AMPARs in OPs normally stimulates OPs to exit the cell cycle and differentiate into OLs, or else enhances the survival of immature OLs once they have formed.

We attempted to distinguish between these two scenarios by comparing the density of Pdgfra$^+$ OPs in $Gria3^{null}2^{-/-}$ and control mice. If OPs continue to divide but are unable to differentiate into OLs in the $Gria3^{null}2^{-/-}$ mice we might expect the density of OPs to increase. We found no significant differences in the density of OPs in $Gria3^{null}2^{-/-}$ SCWM compared to $Gria3^{null}2^{+/+}$ controls (p=0.6 at P3; p=0.85 at P7; p=0.07 at P14; p>0.99 at P21 and p=0.76 at P70, t–test or Mann-Whitney test with Holm-Bonferroni correction; *Figure 7E,F*). This suggests that reduced accumulation of OLs in the SCWM of $Gria3^{null}2^{-/-}$ mice is not caused by a reduction in the rate of differentiation of OPs. However, the reduction in OL density might have resulted from a reduced proliferation rate of OPs – that is, more slowly dividing OPs produce fewer OLs. To assess OP division rates we administered EdU to mice at P3, P7, P14 and P70 (see Materials and methods). The fraction of Pdgfra$^+$ OPs that incorporated EdU ('labelling index') was not detectably different in $Gria3^{null}2^{-/-}$ versus $Gria3^{null}2^{+/+}$ SCWM at any age examined (p=0.69 at P3; p=0.69 at P7; p=0.7 at P14; p=0.34 at P70, Mann-Whitney tests; *Figure 7—figure supplement 1A,B*), nor was there any difference in the fraction of Pdgfra$^+$ OPs in the P14 SCWM that immunolabelled for Ki67, a marker of actively dividing cells (p=0.78, Student's t-test; *Figure 7—figure supplement 1C,D*). These data indicate that the OP division rate is not affected by loss of GluA2 and GluA3.

We performed anti-cleaved Caspase-3 and anti-Olig2 immunolabelling to test directly whether apoptotic death of OL lineage cells was increased by loss of GluA2/GluA3. The proportion of Olig2$^+$ cells that labelled for cleaved Caspase-3 in $Gria3^{null}2^{+/+}$ controls at P14 was 22.7 ± 1.2% (n = 8 mice). In $Gria3^{null}2^{-/-}$ mice the cleaved Caspase-3-positive fraction was increased by a factor ~1.19

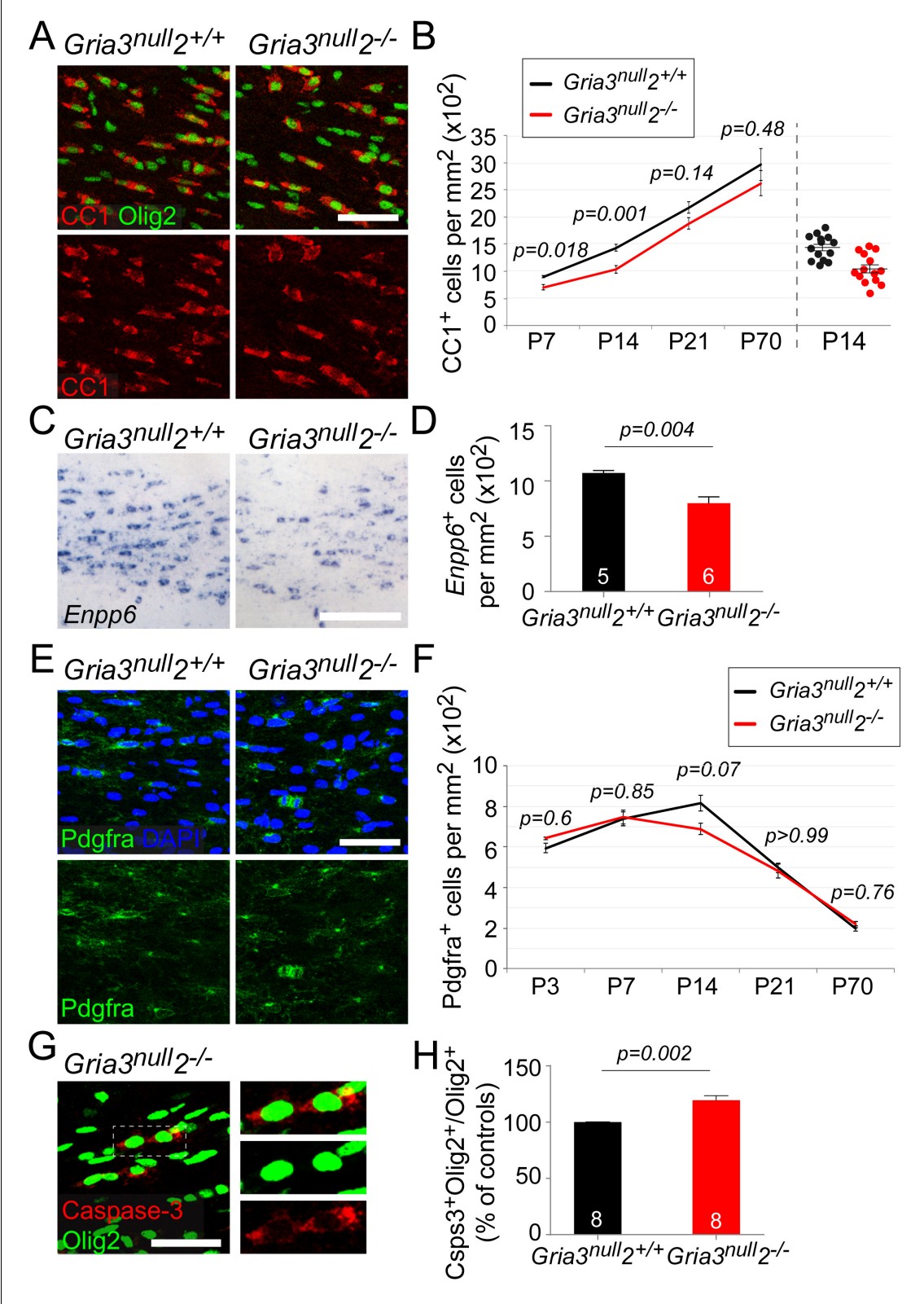

**Figure 7.** *Gria3null2−/−* mice generate fewer OLs in the corpus callosum. All data at P14 unless otherwise stated. (**A**) Immunolabelling for Olig2 and CC1 antigen visualizes differentiated OLs in *Gria3null2+/+* and *Gria3null2−/−* mice. (**B**) The density of CC1+ OLs is significantly less (by ~27%) in *Gria3null2−/−* mice relative to *Gria3null2+/+* controls at both P7 and P14 (t- or Mann-Whitney tests with Holm-Bonferroni correction; n = 7, 14, 7 and 6 mice at P7, P14, P21 and P70, respectively, for *Gria3null2−/−*; n = 5, 14, 9 and 6 mice at P7, P14 and P21 for *Gria3null2+/+*. >800 cells were counted per mouse at all ages).

*Figure 7 continued on next page*

*Figure 7 continued*
Right hand side of the graph visualizes the spread of data at P14. (C) ISH for *Enpp6* in *Gria3^null^2^+/+^* and *Gria3^null^2^−/−^* mice. (D) There is a ~26% reduction in the density of *Enpp6* cells in *Gria3^null^2^−/−^* compared to *Gria3^null^2^+/+^* (p=0.004, Mann-Whitney test). (E) Pdgfra⁺ OPs in *Gria3^null^2^+/+^* and *Gria3^null^2^−/−^* mice. (F) There were no significant differences in the density of Pdgfra⁺ OPs in *Gria3^null^2^−/−^* versus *Gria3^null^2^+/+^* mice between P3-P70, although there was a trend towards transient reduction in cell density at P14 in *Gria3^null^2^−/−^* (t- or Mann-Whitney tests with Holm-Bonferroni correction; n = 4, 8, 14, 7 and 8 mice at P3, P7, P14, P21 and P70, respectively, for *Gria3^null^2^−/−^*, n = 4, 6, 14, 9 and 7 mice at P3, P7, P14, P21 and P70 for *Gria3^null^2^+/+^*. >800 cells were counted per mouse at all ages). (G) Cleaved Caspase-3⁺, Olig2⁺ OL lineage cells in SCWM of P14 *Gria3^null^2^−/−^* mice. Cells in the rectangle (dotted line) are shown on the right at higher magnification. (H) There was a ~19% increase in the fraction of Olig2⁺ cells that expressed cleaved Caspase-3 in *Gria3^null^2^−/−^* compared to *Gria3^null^2^+/+^* littermate controls (p=0.002, Student's t-test with Welch's correction; >800 Olig2⁺ cells were counted in each mouse). Numbers of mice analyzed are indicated in (D) and (H). Scale bars: 50 μm.
The following figure supplements are available for figure 7:
**Figure supplement 1.** The division rate of OPs in the corpus callosum is not affected in *Gria3^null^2^−/−^* mice.
**Figure supplement 2.** mRNA encoding GluA2-4 continue to be expressed in *Enpp6⁺* early-differentiating OLs.

(increased by 19.4 ± 4.2%, p=0.002, Student's t-test with Welch's correction; *Figure 7G,H*.). Published expression databases suggest that AMPAR subunits are expressed in pre-myelinating OLs as well as in OPs, though at a lower level in the former (*Cahoy et al., 2008*; *Zhang et al., 2014*). We confirmed by double ISH for *Enpp6* and *Gria2*, *Gria3* or *Gria4* that some newly-forming *Enpp6⁺* OLs express one or more GluA subunits (*Figure 7—figure supplement 2*). Together, these data indicate that signalling through AMPAR in OL lineage cells promotes survival of newly-differentiating OLs.

## Reduced total myelin in *Gria3^null^2^−/−^* white matter

A reduction in the number of OLs would be expected to lead to a reduction in myelination. To test this, we examined P14 *Gria3^null^2^−/−^* and *Gria3^null^2^+/+^* brains by electron microscopy (EM) and counted the number of transverse myelin figures viewed in parasagittal sections of SCWM (*Figure 8A,B*). This revealed a ~20% reduction in the number of myelin cross-sections in *Gria3^null^2^−/−^* normalized to littermate controls at P14 (p=0.02, Mann-Whitney test), consistent with the ~27% reduction in the number of OLs in *Gria3^null^2^−/−^* mice at the same age (see previous section; *Figure 7*). Therefore, the most parsimonious explanation for the myelin deficit is that the number of OLs is decreased without a compensatory change in the number or length of myelin internodes made by individual OLs. To test this, we dye-filled OLs from P14 *Gria3^null^2^−/−^* and *Gria3^null^2^+/+^* brain slices and measured the lengths and numbers of internodes associated with individual OLs. There was no change in the average length of internodes (p=0.47, Student's t-test) or the average number of internodes per OL (p=0.26, Mann-Whitney test; *Figure 8C,D*). In addition, by EM we found no change in the range or frequency distributions of myelinated (*Figure 8E*) or unmyelinated (*Figure 8F*) axon diameters at P14, arguing against an indirect effect of the *Gria3^null^* and/or *Gria2^−/−^* mutations on OLs and myelin via axons – for example, because the fraction of axons that exceeded the size threshold for myelination might have been reduced in the *Gria* mutants. We also measured myelin thickness, expressed as g-ratio, and found no significant change (p=0.29, Mann-Whitney test; *Figure 8G*).

At P70 there was no longer any significant difference in the number of myelin cross-sections per unit area observed by EM (p=0.57, Mann-Whitney test; *Figure 8B*), consistent with the fact that the numbers of OLs identified by CC1 immunolabelling were similar in *Gria3^null^* and *Gria3^null^2^−/−^* at this age (*Figure 7B*). At P70, as at P14, there was no change in the g-ratios of myelinated axons in *Gria3^null^* versus *Gria3^null^2^−/−^* corpus callosum (p=0.29, Mann-Whitney test; *Figure 8H*). Our data are consistent with a model in which each OL synthesizes a similar total length of myelin sheath in *Gria3^null^2^−/−^* mice as in controls and the overall synthesis of myelin is reduced in proportion to the reduced number of OLs.

## Triple knockout of GluA2/ GluA3/ GluA4 in the OL lineage

Since ~50% of the kainate-evoked current persisted in *Gria3^null^2^−/−^* mice, we questioned whether deletion of all three AMPAR subunits expressed in the OL lineage (GluA2, GluA3 and GluA4) might

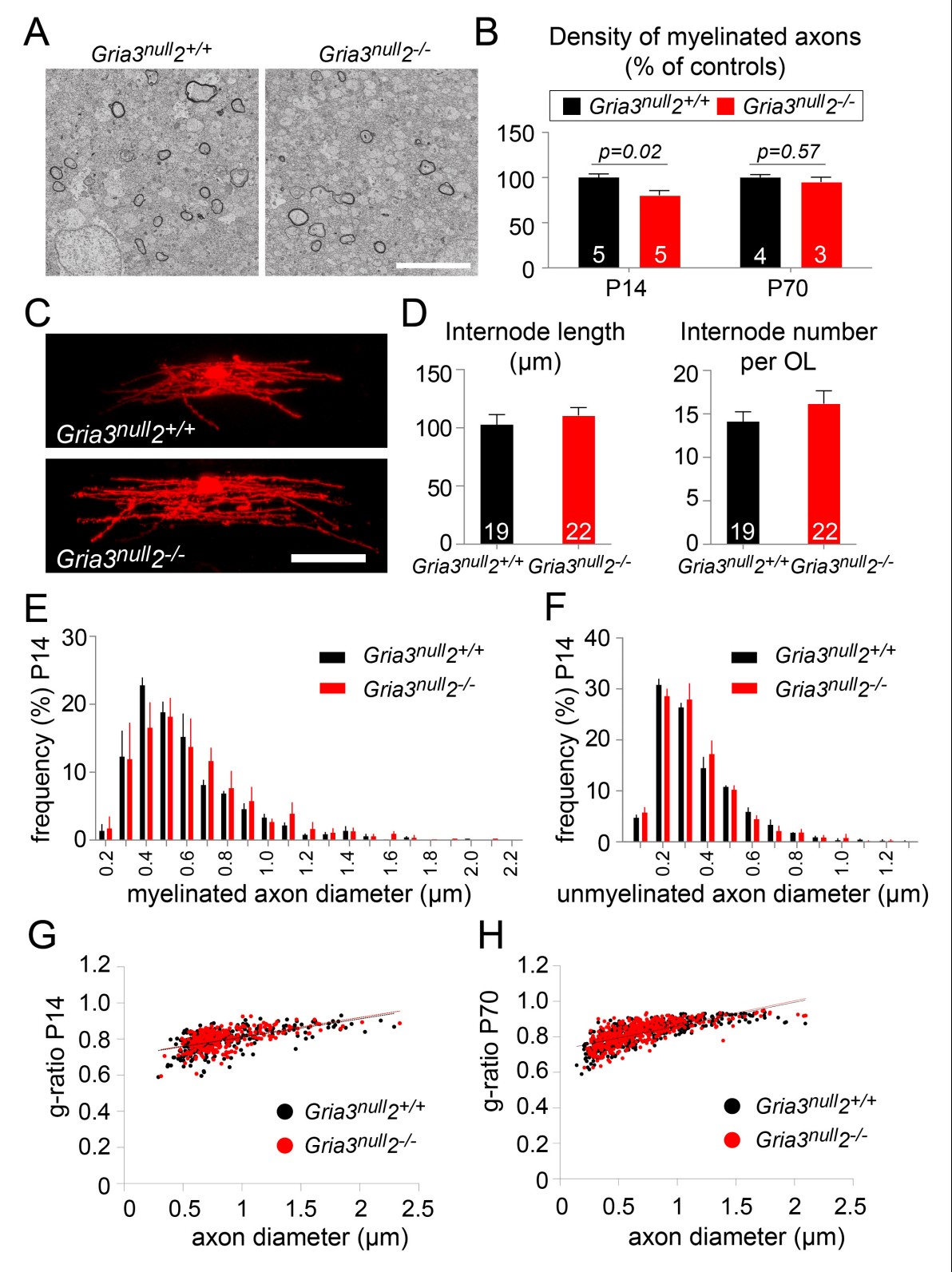

**Figure 8.** Less myelin in *Gria3^null2–/–* corpus callosum. (**A**) Transmission electron micrograph of P14 corpus callosum from *Gria3^null2+/+* and *Gria3^null2–/–* mice. Scale bar: 5 µm. (**B**) There was a ~20% reduction in the number of cross-sectional myelin figures per unit area in *Gria3^null2–/–* compared to *Gria3^null2+/+* mice at P14, but not at P70 (p=0.02 at P14, p=0.57 at P70, Mann-Whitney tests). (**C**) Dye-filled OLs in P14 corpus callosum of *Gria3^null2+/+* and *Gria3^null2–/–* mice. (**D**) There was no difference in the length of the internodes (p=0.47, Student's t-test) or the number of internodes per OL

*Figure 8 continued on next page*

*Figure 8 continued*

(p=0.26, Mann-Whitney test) in *Gria3^{null}2^{–/–}* compared to *Gria3^{null}2^{+/+}* mice. (**E**) Frequency distribution of myelinated axon diameters from P14 *Gria3^{null}2^{+/+}* and *Gria3^{null}2^{–/–}* mice (100–350 axons measured in each of three mice per group). (**F**) Frequency distribution of unmyelinated axon diameters from P14 *Gria3^{null}2^{+/+}* and *Gria3^{null}2^{–/–}* mice (>300 axons measured in each of three mice per group). (**G**) Scatter plots of g-ratios as a function of axon diameter in P14 *Gria3^{null}2^{+/+}* and *Gria3^{null}2^{–/–}* mice (>260 axons measured from four mice per group). (**H**) Scatter plots of g-ratios as a function of axon diameter in P70 *Gria3^{null}2^{+/+}* and *Gria3^{null}2^{–/–}* mice (>150 axons measured from each of three mice per group). Number of mice analyzed is indicated in (**B**) and number of cells analyzed in (**D**).

result in a more severe phenotype. We therefore generated triple-KO mice with the genotype *Sox10-Cre: Gria3^{null}: Gria2^{flox/flox}: Gria4^{flox/flox}: Rosa-YFP* (referred to as *Gria3^{null}2^{–/–}4^{–/–}*) (***Figure 4—figure supplement 1C***). Littermate controls for assessment of cell densities and myelination were *Gria3^{null}* with two functional copies of *Gria2* and *Gria4* because they did not carry the *Sox10-Cre* transgene. For electrophysiological experiments that used *YFP* fluorescence for OP identification (requiring both *Sox10-Cre* and *Rosa-YFP* transgenes), *Gria3^{null}2^{–/–}* littermates carrying wild type alleles of *Gria4* were used for comparison (***Figure 4—figure supplement 1C***).

Whole-cell patch-clamp recordings of YFP^+ OPs in the P14 SCWM revealed that the current density of kainate-evoked current elicited by application of 100 µM kainate at –63 mV was dramatically reduced in *Gria3^{null}2^{–/–}4^{–/–}* to ~23% of that observed in *Gria3^{null}2^{–/–}* littermates (p<10^{–4}, Student's t-test with Welch correction; ***Figure 9A,B***), suggesting that very few, if any, AMPARs are present in *Gria3^{null}2^{–/–}4^{–/–}* OPs (the kainate-evoked current in *Gria3^{null}2^{–/–}4^{–/–}* OPs is ~13% of that in *Gria3^{null}* OPs; compare ***Figures 5B*** and ***9B***). In support of this, application of 50 µM GYKI blocked the kainate-evoked current in *Gria3^{null}2^{–/–}4^{–/–}* only by ~39% as opposed to ~80% in *Gria3^{null}2^{–/–}* (p=0.005, Mann-Whitney test; ***Figure 9C***). Moreover, the I-V relationship of the kainate-evoked current was nearly linear in *Gria3^{null}2^{–/–}4^{–/–}* OPs, not inwardly-rectifying as in *Gria3^{null}2^{–/–}* (different rectification indices, p=0.005, Mann-Whitney test; ***Figure 9D,E***). This suggests that receptors other than AMPAR – most likely kainate receptors (KAR) – mediate the majority of the (much reduced) kainate-evoked response in *Gria3^{null}2^{–/–}4^{–/–}* OPs. Consistent with the apparent lack of AMPAR in *Gria3^{null}2^{–/–}4^{–/–}* OPs, and with AMPARs rather than KARs being the main receptor type activated at axon-OP synapses, almost no EPSCs were detected following application of 100 µM RR to brain slices (significantly less than in *Gria3^{null}2^{–/–}*; p=0.03, Mann-Whitney test) (***Figure 9F,G***). These data show that AMPAR signalling is essentially abolished in *Gria3^{null}2^{–/–}4^{–/–}* OPs.

We counted CC1^+ OLs and Pdgfra^+ OPs in the SCWM of *Gria3^{null}2^{–/–}4^{–/–}* mice at P14 and P53. At P14 there was a ~22% reduction in the density of OLs relative to *Gria3^{null}* controls and at p53 a ~26% reduction (p=0.026 at P14; p=0.029 at P53, Mann-Whitney test, ***Figure 10A,B***) without a concomitant change in the density of OPs (p=0.11 at P14; p=0.7 at P53, Mann-Whitney test) (***Figure 10C,D***). The fraction of Olig2^+ cells that expressed cleaved Caspase-3 in SCWM of P14 *Gria3^{null}* controls was 24.8 ± 0.8% (n = 5 mice) and this increased in *Gria3^{null}2^{–/–}4^{–/–}* mice by a factor of ~1.24 (increased by 24.2 ± 4.8%, p=0.004, Mann-Whitney test; ***Figure 10E,F***). As observed for *Gria3^{null}2^{–/–}* mice, there was no significant change in the average length (p=0.93, Student's t-test) or number (p=0.41, Student's t-test) of internodes per individual *Gria3^{null}2^{–/–}4^{–/–}* OL, analyzed by dye-filling OLs in brain slices (***Figure 10G,H***). Taken together, our data demonstrate that signalling through AMPAR stimulates OL production and myelination in the white matter by enhancing survival of newly-formed OLs, without a detectable effect on OL morphology.

## Discussion

Previous studies have shown that signalling between electrically active axons and OL lineage cells can influence several aspects of OL development, including proliferation and differentiation of OPs and the initiation and stabilization of myelin internodes. We have now provided the first direct evidence that synaptic signalling through AMPARs in OPs and/or early OLs modulates OL development in vivo. Knock-out of *Gria2* or *Gria3* alone had no discernible effect on OL development but double-KO of *Gria2/3* in the OL lineage reduced, and triple-KO of *Gria2/3/4* abolished, AMPAR-mediated signalling in OPs and led to reduced accumulation of mature OLs and myelin without affecting the proliferation or number of OPs. The presence of GluA2 subunits (encoded by *Gria2*) is known to

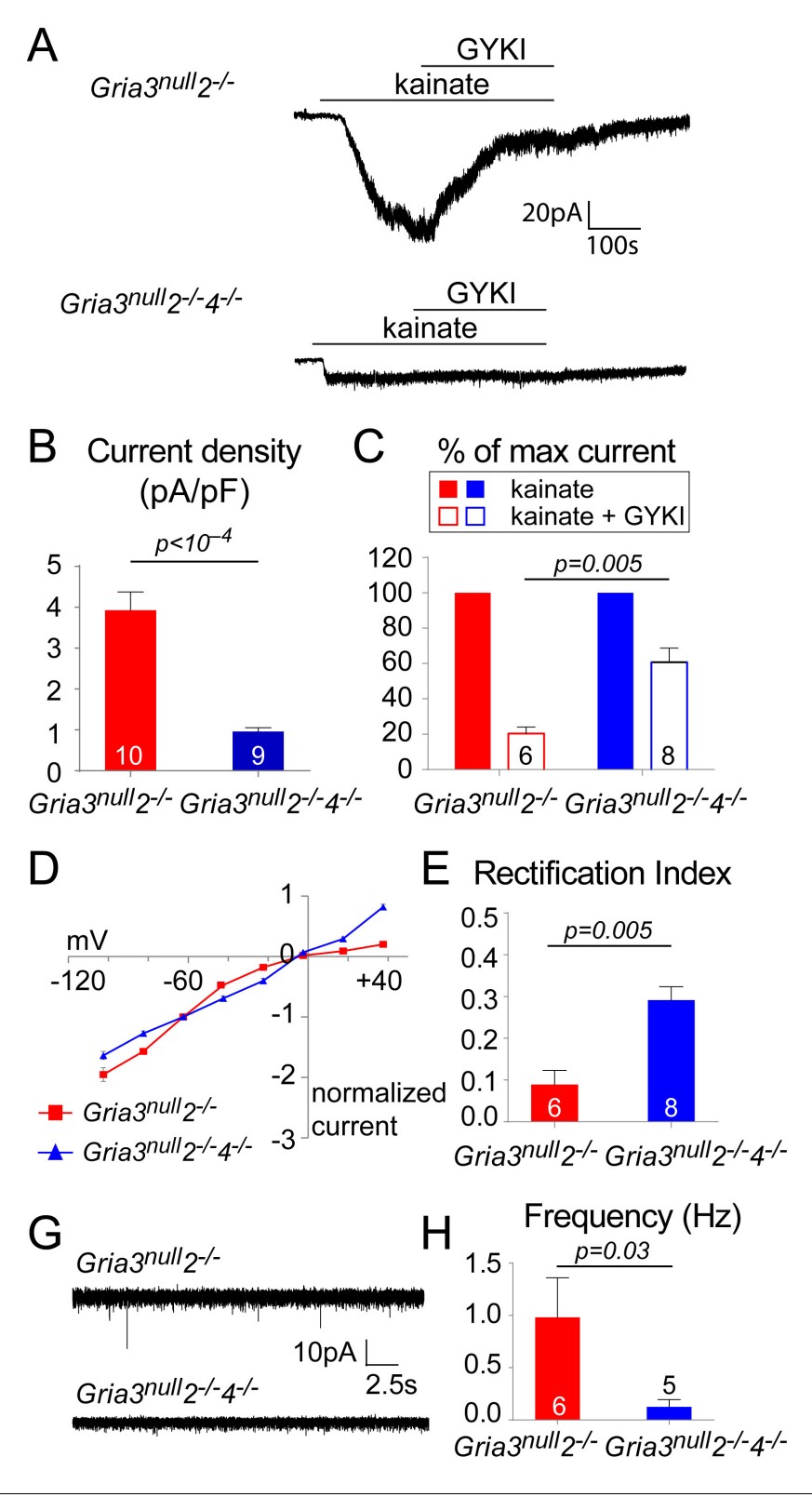

**Figure 9.** $Gria3^{null}2^{-/-}4^{-/-}$ mice lack AMPAR-mediated inputs. (**A**) The kainate-evoked current is greatly reduced in OPs of P14 $Gria3^{null}2^{-/-}4^{-/-}$ mice compared to $Gria3^{null}2^{-/-}$ littermates and is less effectively blocked by the AMPAR antagonist GYKI-52466 (GYKI, 50 µM). (**B**) Mean kainate-evoked current density in P14 $Gria3^{null}2^{-/-}4^{-/-}$ OPs is ~24% that of $Gria3^{null}2^{-/-}$ littermates ($p<10^{-4}$, Student's t-test with Welch correction), corresponding to ~13% that of $Gria3^{null}$ OPs (**Figure 5B**). (**C**) GYKI blocks the kainate-evoked current by ~80% in $Gria3^{null}2^{-/-}$, but only by ~39% in. $Gria3^{null}2^{-/-}4^{-/-}$ mice

*Figure 9 continued on next page*

*Figure 9 continued*

(p=0.005, Mann-Whitney test). (D) Current-voltage plot showing inward rectification in OPs of P14 *Gria3^null^2^−/−^* mice but not *Gria3^null^2^−/−^4^−/−^*, when 100 μM spermine is included in the patch pipette. (E) Rectification indices of cells recorded in (D) are significantly increased in *Gria3^null^2^−/−^4^−/−^* compared to *Gria3^null^2^−/−^* littermates (p=0.005, Mann-Whitney test). (F) EPSCs evoked in OPs by Ruthenium Red (RR, 100 μM) in the corpus callosum of P14 *Gria3^null^2^−/−^* and *Gria3^null^2^−/−^4^−/−^* mice. (G) The frequency of RR-evoked EPSCs is dramatically reduced in *Gria3^null^2^−/−^4^−/−^* compared to *Gria3^null^2^−/−^* littermates (p=0.03, Mann-Whitney test). Numbers of cells analyzed are indicated in (B), (C), (E) and (G).

suppress the $Ca^{2+}$ permeability of AMPARs (provided GluA2 is the Q –> R edited isoform). The fact that knockout of *Gria2* alone did not affect OL development implies that the reduction in OL number observed in the *Gria2/3* and *Gria2/3/4* mutants results from an aspect of AMPAR signalling other than $Ca^{2+}$ permeability, for example the induced depolarization or the $Na^+$ influx that it produces. Thus, AMPAR signalling in OPs and/or early-differentiating OLs is a positive regulator of OL development; we speculate that this favours myelination of active axons that are releasing glutamate, thus accelerating development of a functioning CNS.

## Effect of AMPAR subunit knockouts on synaptic input to OPs

OPs in developing white matter receive glutamatergic synaptic input from unmyelinated axons in white matter. Most glutamatergic synaptic current in OPs is via AMPAR (*Lin and Bergles, 2004*; reviewed by *Bergles and Richardson, 2015*). We have shown that OL lineage cells in the SCWM express mRNA encoding AMPAR subunits GluA2, GluA3 and GluA4 but not GluA1 - the same spectrum of AMPAR subunits reported for rat and mouse OPs in culture (*Patneau et al., 1994*; *Zhang et al., 2014*). Knocking out GluA2 and GluA3 in OPs reduced the AMPAR-mediated current to ~53% of normal, while knocking out all three subunits GluA2/3/4 reduced it to ~13% of normal. Surprisingly, we found that in OPs lacking GluA2 and GluA3 the amplitude of RR-induced synaptic events was unchanged while their frequency was reduced to <30% of normal, while in OPs lacking all AMPAR subunits their frequency was reduced to only ~1% of control. The simplest interpretation is that there are fewer AMPAR-containing synapses in double-KO OPs (and almost none in triple-KOs), but that those synapses that survive contain a normal number of AMPARs. In support of this it has been shown in neurons that, when the number of post-synaptic AMPARs is increased experimentally, an increase in frequency of EPSPs with no change in their amplitude is observed (*Sinnen et al., 2017*), suggesting that the number of functioning synaptic connections is determined, at least in part, by the number of postsynaptic AMPARs available, and that quantal amplitude is regulated independently of the number of synaptic connections. In the GluA2/3 double-KO the surviving synapses must contain exclusively or predominantly GluA4 subunits, which are known to form functional homo-tetramers (*Keinänen et al., 1990*). We cannot tell whether separate 'GluA4-only' and 'GluA4-lacking' synapses are normally present on OPs, although distinct sets of GluA2/GluA3-containing and GluA2/GluA3-lacking synapses have been described for neurons in the carp retina (*Schultz et al., 2001*). In any case our data provide striking genetic confirmation that almost all glutamatergic synaptic input to OPs is via AMPARs composed of GluA2, GluA3 and GluA4 subunits.

## AMPAR signalling does not affect OP proliferation

We found that OP proliferation and OP number were normal in *Gria3^null^2^−/−^* and *Gria3^null^2^−/−^4^−/−^* mice. Previous studies disagree over whether activity-driven glutamate release stimulates, inhibits or has no effect on proliferation of OPs. Experiments with organotypic rat or mouse cerebellar slice cultures found that adding AMPA/ kainate receptor agonists inhibited proliferation of OPs (*Yuan et al., 1998*; *Fannon et al., 2015*). However, direct electrical stimulation of glutamatergic pyramidal neurons in the rat motor cortex was found to stimulate OP proliferation in the cerebro-spinal tract (*Li et al., 2010*), and optogenetic stimulation of projection neurons in mouse premotor cortex stimulated OP proliferation in the premotor pathway, including the SCWM (*Gibson et al., 2014*). Moreover, blocking the activity of retinal ganglion cells (RGCs) and their axons in the mouse optic nerve by intra-ocular injection of tetrodotoxin inhibited OP proliferation (*Barres and Raff, 1993*). On the other hand, sensory deprivation of mouse barrel cortex by whisker removal (*Mangin et al., 2012*; *Hill et al., 2014*), which presumably restricts glutamate release in the cortex, increased OP proliferation in the barrels, in keeping with the data from brain slice cultures (*Yuan et al., 1998*;

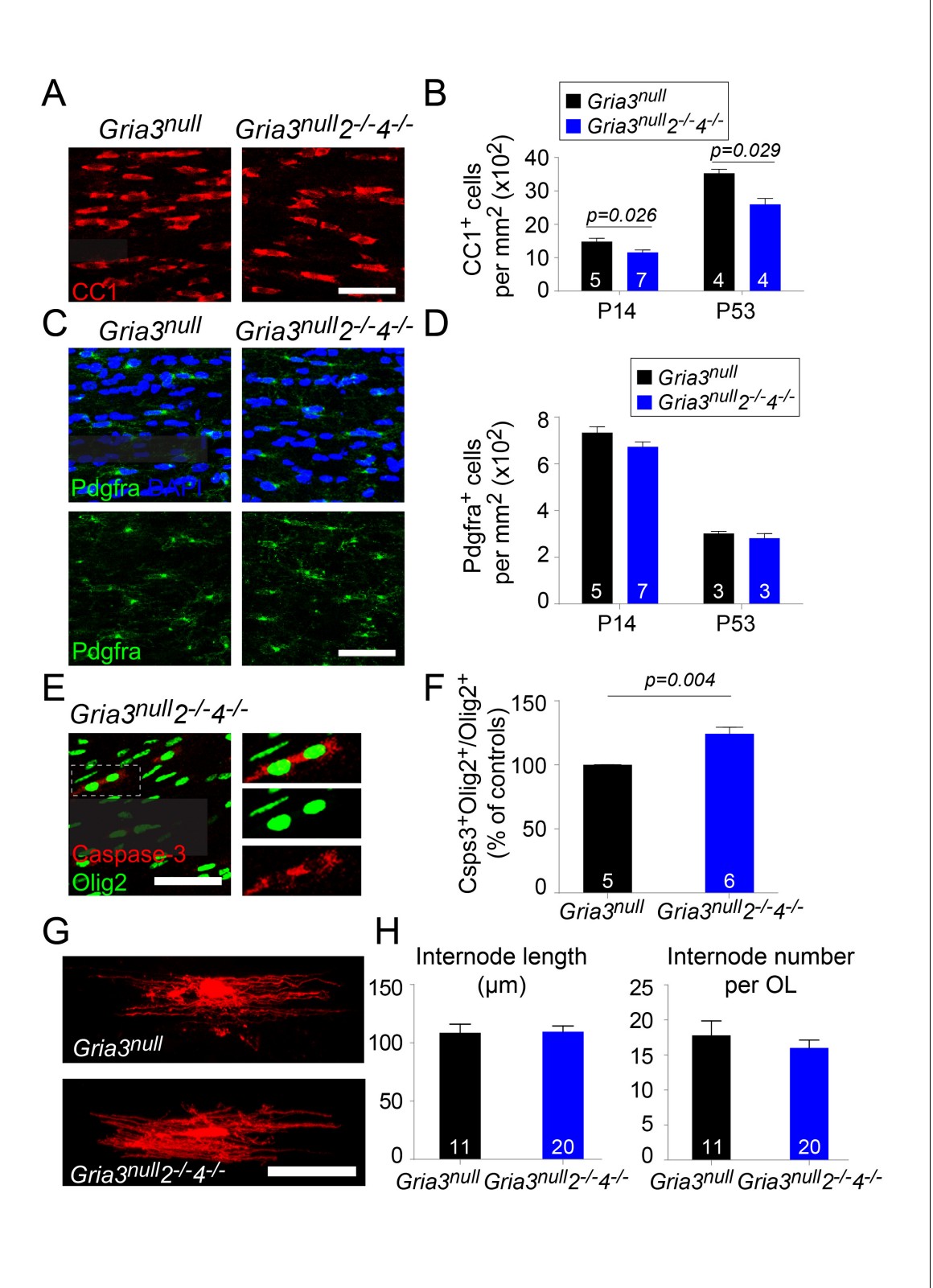

**Figure 10.** *Gria3^null^2^−/−^4^−/−* mice generate fewer OLs in the corpus callosum. (**A**) CC1 immunolabelling marks differentiated OLs in *Gria3^null^* and *Gria3^null^2^−/−^4^−/−* mice. (**B**) The density of CC1+ OLs is significantly less (by ~22% at P14 and ~26% at P53) in *Gria3^null^2^−/−^4^−/−* mice relative to *Gria3^null^* controls (p=0.026 at P14, p=0.029 at P53, Mann-Whitney test; >900 cells were counted per mouse at all ages). (**C**) Pdgfra+ OPs in *Gria3^null^* and *Gria3^null^2^−/−^4^−/−* mice. (**D**) There were no significant differences in the density of Pdgfra+ OPs at P14 or P53 in *Gria3^null^2^−/−^4^−/−* versus *Gria3^null^* mice

*Figure 10 continued on next page*

*Figure 10 continued*

(p=0.11 at P14; p=0.7 at P53, Mann-Whitney test; >400 cells were counted per mouse at all ages). (**E**) Cleaved Caspase-3$^+$, Olig2$^+$ OL lineage cells in *Gria3$^{null}$2$^{-/-}$4$^{-/-}$* mice. Cells in the rectangle (dotted line) are shown on the right at higher magnification. (**F**) There was a ~24% increase in the fraction of Olig2$^+$ cells that expressed cleaved Caspase-3 in *Gria3$^{null}$2$^{-/-}$4$^{-/-}$* compared to *Gria3$^{null}$* mice (p=0.004, Mann-Whitney test; >1500 Olig2$^+$ cells were counted in each mouse). (**G**) Dye-filled OLs in P14 corpus callosum of *Gria3$^{null}$* and *Gria3$^{null}$2$^{-/-}$4$^{-/-}$* mice. (**H**) There was no change in the length of the internodes (p=0.93, Student's t-test) or the number of internodes per OL (p=0.41, Student's t-test) in *Gria3$^{null}$2$^{-/-}$4$^{-/-}$* compared to *Gria3$^{null}$* mice. Numbers of mice analyzed are indicated in (**B**), (**D**) and (**F**) and numbers of cells in (**H**). Scale bars: 50 µm.

*Fannon et al., 2015*). Against all of the above, a recent in vivo study, in which glutamate release from the axons of mouse RGCs was genetically attenuated, found no effect on OP proliferation in the optic tract (*Etxeberria et al., 2016*). Genetic block of vesicular release from longitudinally-projecting axons in the zebrafish spinal cord also had no effect on OP proliferation (*Hines el al., 2015*; *Mensch et al., 2015*), although numbers of OL lineage cells were reduced by ~10%, apparently because of reduced specification of OPs from their stem cell precursors in the embryonic ventricular zone (*Mensch et al., 2015*). It is unlikely that we would have detected an effect of this size, even assuming it would not have been overridden by homeostatic controls on cell number in the perinatal mouse brain.

The data cited above are contradictory and confusing. However, the experiments are not directly comparable to each other. For example, pharmacological manipulation of slice cultures is not targeted to specific cells and opens the possibility that there might be indirect effects mediated via neurons, some of which also express AMPA/ kainate receptors. The same reservation might apply to some of the in vivo experiments cited above – on top of which, stimulating or inhibiting electrical activity or vesicular release in vivo is likely to affect more than just AMPAR-mediated signalling. For example, electrically active neurons release Neuroligin-3 (NLGN3), a mitogen for malignant glioma cells, which are believed to be derived from and closely related to OPs (*Venkatesh et al., 2015*). Our conditional knockout approach is tightly focused on AMPAR-mediated effects in OL lineage cells and avoids confounding influences by other signalling pathways, direct or indirect. We conclude, as discussed below, that signalling through AMPAR in OL lineage cells promotes survival of newly-differentiating OLs.

## AMPAR signalling promotes the survival of differentiating OLs

Reduced AMPAR signalling in the SCWM of our *Gria3$^{null}$2$^{-/-}$* mice resulted in a modest (~27%) reduction in OL accumulation in the early postnatal period (around P14), but OL numbers recovered to normal levels in adult mice. Additional deletion of *Gria4* did not reduce the number of OLs further, but the reduction now persisted into adulthood. We assume that these changes reflect the loss of synaptic currents that we have documented above. The relatively modest effect on OL dynamics of ablating AMPAR might reflect the fact that glutamate can signal to OPs through receptors other than AMPAR (including NMDAR, KAR or mGluR), and that glutamate is only one of a number of signalling molecules that communicate axonal activity to OPs.

Previous studies into the effects of glutamatergic signalling on OL differentiation have been conflicting. Experiments in dissociated cell cultures and slice cultures from rat or mouse indicated that blocking AMPA/ kainate receptors promoted OL differentiation (*Gallo et al., 1996*; *Yuan et al., 1998*; *Fannon et al., 2015*) while preventing their further maturation and expression of myelin proteins (*Fannon et al., 2015*). In vivo, manipulating electrical activity pharmacologically (*Demerens et al., 1996*), electrically (*Li et al., 2010*), optogenetically (*Gibson et al., 2014*) or physiologically (*Simon et al., 2011*; *McKenzie et al., 2014*; *Xiao et al., 2016*) have suggested that axonal activity promotes OL generation and myelin formation in rats and mice. However, inhibiting vesicular release from axons in the developing zebrafish spinal cord by pan-neuronal expression of tetanus toxin had no effect on OL differentiation, but reduced the number and length of myelin sheaths synthesized by individual OLs (*Hines et al., 2015*; *Mensch et al., 2015*). This effect was observed for OLs that myelinated some but not all classes of axon (*Koudelka et al., 2016*) – consistent with the idea that there are both activity-dependent and –independent mechanisms of myelination (*Lundgaard et al., 2013*). However, tetanus toxin inhibits all SNARE-dependent vesicular release, including release of neurotransmitters other than glutamate (e.g. ATP and GABA) that have also

been implicated in the control of myelination (*Stevens et al., 2002*; *Ishibashi et al., 2006*; *Zonouzi et al., 2015*; *Hamilton et al., 2017*). Further complicating the picture, inhibition of glutamate release from the axons of RGCs increased OL number in the mouse optic tract (*Etxeberria et al., 2016*). Although the latter study preferentially affected glutamate signalling it did not separate AMPAR-mediated effects from those mediated by e.g. NMDAR or mGluR.

We found that the reduced number of differentiated OLs in the double- and triple-KOs was a result of increased death of OL lineage cells – most likely newly-formed, pre-myelinating OLs. In keeping with this, we found that *Gria2-4* continue to be expressed in $Enpp6^+$ early-differentiating OLs, suggesting that some AMPAR-mediated signalling can still occur during the early stages of OL differentiation, even though AMPAR-mediated EPSCs are down-regulated in maturing OLs (*Kukley et al., 2010*; *De Biase et al., 2010*). Synaptic input – and activation of AMPARs in particular – has been shown to promote neuronal survival during early postnatal development (*Heck et al., 2008*) and a recent study showed that sensory input from the whiskers to somatosensory barrel cortex promotes the survival of newly-differentiating OLs (*Hill et al., 2014*). Thus, AMPAR signalling in OL lineage cells could be one of the mechanisms ensuring that only pre-myelinating OLs that make contact with active axons survive to myelinate in the longer term (*Barres et al., 1992*, *1993*; *Trapp et al., 1997*).

Electrical activity is also known to affect the terminal stages of OL differentiation, that is, myelin synthesis (*Demerens et al., 1996*; *Wake et al., 2011*; *Lundgaard et al., 2013*; *Gibson et al., 2014*; *Wake et al., 2015*). Glutamate receptors other than AMPAR, that is, NMDAR and mGluR, have been suggested to regulate this process (*Wake et al., 2011*; *Lundgaard et al., 2013*), possibly by regulating the energy supply to developing OLs (*Saab et al., 2016*) – although a major effect of NMDARs in vivo has been disputed (*De Biase et al., 2011*; *Guo et al., 2012*). In our present study we found that the amount of myelin in the SCWM of $Gria3^{null}2^{-/-}$ mice was reduced in proportion to the reduced number of OLs. The g-ratio, as well as the internode number and length, were unchanged, suggesting that once a pre-myelinating OL makes a commitment to myelinate an axon, the process continues until a normal internode number and length, and a normal number of myelin wraps are achieved. Therefore, the activity-dependent regulation of internode length that has been observed in vitro (*Wake et al., 2015*) and in vivo (*Koudelka et al., 2016*) is presumably mediated by glutamate receptors other than AMPAR, or by signals other than glutamate. In short, we have shown that AMPAR signalling stimulates myelin formation in mice by increasing the number of myelinating OLs that develop from OPs, not by regulating myelin synthesis per se.

In summary, our study demonstrates that AMPAR signalling in OL lineage cells promotes OL development and myelination during postnatal development. Elucidating further mechanisms of signalling from neurons to OL lineage cells will be crucial for understanding developmental myelination and for devising therapies to promote remyelination in diseases of the white matter.

## Materials and methods

### Transgenic mice

AMPAR subunits GluA1-4 are encoded by separate genes *Gria1-4*. In the $Gria2^{flox/flox}$ line, exon 11 of *Gria2* is flanked by *loxP* sites (*Shimshek et al., 2006a*). $Gria4^{flox/flox}$ mice (*Gria4tm1Mony*; MGI:3798453) also have exon 11 flanked by *loxP* sites (*Fuchs et al., 2007*). *Gria3* null mice ($Gria3^{-/-}$ or $Gria3^{-/Y}$) have exon 11 deleted in the germ line (*Sanchis-Segura et al., 2006*). *Rosa-YFP* reporter mice (*Srinivas et al., 2001*) allow Cre-induced expression of the enhanced yellow fluorescent protein (YFP). Most transgenic lines were maintained as homozygous breeding colonies. *Sox10-Cre* mice (line 22–7) (*Matsuoka et al., 2005*) express Cre recombinase under *Sox10* transcriptional control in a phage artificial chromosome (PAC). They were maintained as a heterozygous breeding colony and only female carriers were used for experiments, because *Sox10-Cre* is expressed and causes recombination in the male germ line. Our breeding schemes are shown in *Figure 4—figure supplement 1A–C*. Mice were maintained on a 12 hr light-dark cycle. The day of birth was designated postnatal day 0 (P0). For histological experiments the age of the animals is stated; for electrophysiological experiments a range of ages P13-P16 was used, referred to collectively as P14.

## Electrophysiology

Coronal slices (225 µm thick) including the corpus callosum were prepared from the forebrain of P13-15 mice. Sections were prepared in ice-cold oxygenated solution containing 124 mM NaCl, 26 mM NaHCO$_3$, 2.5 mM KCl, 1 mM NaH$_2$PO$_4$, 2.5 mM CaCl$_2$, 2 mM MgCl$_2$, 10 mM D-glucose, 1 mM Na-kynurenate, pH 7.4, bubbled with a 95% O$_2$/5% CO$_2$ mixture. After sectioning, slices were maintained in the same solution at 20–25°C before use. Brain slices were superfused with either HEPES-buffered solution containing 140 mM NaCl, 2.5 mM KCl, 2 mM CaCl$_2$, 1 mM MgCl$_2$, 10 mM HEPES, 1 mM NaH$_2$PO$_4$, and 10 mM D-glucose, pH 7.4 (adjusted with NaOH), bubbled with 100% O$_2$, or with bicarbonate-buffered solution containing 124 mM NaCl, 26 mM NaHCO$_3$, 2.5 mM KCl, 1 mM NaH$_2$PO$_4$, 2.5 mM CaCl$_2$, 1 mM MgCl$_2$, 10 mM D-glucose, pH 7.4, bubbled with 95% O$_2$/5% CO$_2$. For Ruthenium Red (RR) –mediated potentiation of the frequency of excitatory postsynaptic current (EPSCs) we used bicarbonate-buffered solution; for other experiments we used HEPES-buffered solution.

Patch pipettes were pulled from thick-walled borosilicate glass capillaries (type GC150F-10; Harvard Apparatus) to a resistance of 3–5 MΩ. For RR experiments the intracellular solution contained 140 mM CsCl, 10 mM HEPES, 2 mM MgATP, 0.5 mM CaCl$_2$, 4 mM NaCl, 5 mM EGTA, 0.052 mM Alexa Fluor 568, pH 7.3 (adjusted with CsOH). For the rest of the experiments the intracellular solution contained CsCl 145 mM, 10 mM HEPES, 4 mM MgATP, 2.5 mM NaCl, 1 mM EGTA, 0.052 mM Alexa Fluor 568, pH 7.2 (adjusted with CsOH). When indicated, spermine (Sigma) was added to the intracellular solution at a final concentration of 100 µM. For both solutions the junction potential of –3 mV has been corrected for.

YFP-expressing OPs in the corpus callosum were selected for whole-cell patch-clamping based on their size and shape in the fluorescence microscope, and their membrane properties. Cells with voltage-gated Na$^+$ currents, ≥1 GΩ membrane resistance and ≤35 pF capacitance were considered to be OPs. Due to the high membrane resistance of OPs, recordings were made without series resistance compensation. In some experiments the following drugs were applied by adding to the superfusing extracellular solution: kainic acid monohydrate (100 µM; Sigma) to activate AMPAR and kainate receptors, GYKI52466 (GYKI; 50 µM; Sigma) to block AMPARs, or RR (100 µM; Sigma) to evoke transmitter release. To record the OP's voltage-gated currents and the I–V relationship of drug-evoked currents, 20 mV voltage steps of 200 ms duration were applied from a holding potential of –63 mV (voltage range from –103 mV to +17 mV). For RR-evoked EPSCs the holding potential was –83 mV to increase the driving force for excitatory synaptic events and thus increase the chance of detecting the events.

For evoked synaptic responses cells were clamped at –63 mV and the responses were elicited by stimulation (Digitimer Limited) of callosal axons using a glass micropipette filled with external solution (resistance 1–2 MΩ) placed ∼150–200 µm away from the recorded cell in the corpus callosum. For minimal stimulation, activating only one axon with input to the recorded cell, the stimulus strength was gradually reduced until no post-synaptic current was detected in the patch-clamped OP. Then the stimulus was gradually increased until a post-synaptic response was detected. In each cell 50–80 trials under these conditions were recorded and analyzed. The amplitude of the unitary EPSC was obtained by averaging those events where an EPSC was observed. For estimation of paired-pulse ratio (PPR), two strong (90 V, 800 µs) stimulation pulses, 25 ms apart, were delivered ten times, responses were averaged and the PPR was calculated by dividing the peak current generated by the second pulse by the peak current generated by the first pulse.

Responses were recorded using an Axopatch 200B amplifier with pClamp 9 (Axon Instruments) and Axoscope 9 (Axon Instruments) software. The majority of membrane currents were filtered at 5–10 kHz, except for spontaneous synaptic currents, which were filtered at 1 kHz. The sampling rate was always at least three times the filtering frequency.

Data were analyzed off-line using Clampfit 9 (Axon Instruments). Membrane resistance and capacitance were estimated by fitting the transient current elicited by a 5 mV voltage step to a single exponential. Series resistance was monitored and was typically 10–20 MΩ. Cells were excluded from the analysis when the resistance changed by >50% during the recording and/or was >30 MΩ (unless otherwise stated). The amplitude of voltage-gated Na$^+$ currents was calculated by subtracting the passive components of current responses evoked by voltage steps. In particular, the current response to a 20 mV hyperpolarizing voltage step with no active components, comprising the

capacitive current and an ohmic leak conductance (which are proportional to the size of the voltage step) was linearly scaled according to the size of the voltage step applied. This linearly scaled passive component was then subtracted from all of the voltage-gated $Na^+$ current responses evoked by depolarizing voltage steps.

For detection of RR-evoked EPSCs, traces were high pass filtered at 2 Hz to eliminate drift in the baseline over time. The events were automatically detected using a threshold search with the following criteria: amplitude threshold set at three times the s.d. of the noise in regions of the trace lacking obvious EPSCs, minimum event duration set to 1 ms and rising phase faster than the decay. The detected events were then either accepted or rejected by the experimenter. The frequency was calculated for each minute of drug application. For the majority of cells, the maximum frequency was reached after 3 min of RR application. Therefore, the frequency of the events recorded between 3 and 6 min after application of the drug was used for comparison between $Gria3^{null}2^{-/-}$ and control mice. Event frequency during GYKI application was estimated over the last 3 min of drug application.

To quantify kainate-evoked currents, the peak kainate-evoked current was measured at a nominal holding potential of –63 mV. Kainate-evoked current magnitudes were then corrected after the experiment for voltage errors generated by series resistance, using the value of Rs, a reversal potential of 0 mV and the current flowing before and after applying kainate. All cells, irrespective of their series resistance, were included in this analysis.

Numbers of cells analyzed are indicated in the main text. For most experiments, cells from more than three mice of a given genotype were analysed, apart from the PPR experiment (*Figure 6F*), the $Gria3^{null}2^{+/+}$ group in the experiments of *Figure 5B,C* and the experiments in *Figure 9* where two animals were used. Only one cell was recorded from in each slice, so the numbers of cells given are also the numbers of slices.

## Internode measurements of dye-filled OLs

Coronal slices (225 µm thick) of P13-P15 mouse forebrain were prepared as described above (section on *Electrophysiology*). The slices were superfused with bicarbonate-buffered solution containing 124 mM NaCl, 2.5 mM KCl, 26 mM $NaHCO_3$, 1 mM $NaH_2PO_4$, 2 $CaCl_2$, 1 mM $MgCl_2$, 10 mM glucose, pH 7.14, bubbled with a 95% $O_2$/5% $CO_2$ mixture. Mature OLs in the corpus callosum were identified by their location and morphology as revealed by expression of YFP. The cells were whole-cell voltage clamped at –75 mV for 8 min to allow dye to enter the internodes fully. The pipettes had a series resistance of 8–35 MΩ and the internal solution was 145 mM CsCl, 10 mM HEPES, 4 mM MgATP, 2.5 mM NaCl, 1 mM EGTA and 0.5 mM Alexa Fluor 594, adjusted to pH 7.4 with CsOH. After dye filling, the slices were immediately fixed in 4% PFA for 1–4 hr, washed three times in PBS, mounted on slides under coverslips and imaged in a Zeiss LSM700 confocal microscope. Confocal stacks (between 17 and 80 images separated by 1 µm steps) were made of each OL and internode lengths were measured in three dimensions using the Fiji (ImageJ) simple neurite tracer.

## In vivo EdU labelling

5'-ethynyl-2'-deoxyuridine (EdU, Invitrogen, 5 mg/ml in phosphate-buffered saline) was administered to P3 mice by a single 30 µl subcutaneous injection (in the scruff of the neck) and the mice were perfused 4 hr later. P7 mice were injected subcutaneously twice with 60 µl of EdU solution, 3 hr apart, and perfused 3 hr after the second injection. EdU was administered to P14 mice by intra-peritoneal injection (5 mg per kg body weight) 3 times at 2.5 hr intervals and the mice were perfused 2.5 hr after the final injection. EdU was provided to P70 mice via their drinking water (0.2 mg/ml EdU) with free access for 6 days. These regimens were chosen in an attempt to label a comparable proportion of OPs at each age, given that the cell cycle time is known to increase from less than a day to >10 days over this age range (*Rivers et al., 2008*; *Psachoulia et al., 2009*; *Young et al., 2013*).

## Tissue processing

Mice were intracardially perfusion-fixed with 4% (w/v) paraformaldehyde (PFA; Sigma) in PBS. Brain tissue was dissected and post- fixed in 4% PFA overnight at 4°C. Tissue was cryoprotected in diethylpyrocarbonate (DEPC)-treated 20% (w/v) sucrose (Sigma) for 24–48 hr in PBS. It was

then frozen in Optimal Cutting Temperature (OCT) medium (Tissue Tek) on the surface of dry ice and stored at –80°C until needed.

## In situ hybridization (ISH)

ISH was as described at http://www.ucl.ac.uk/~ucbzwdr/In%20Situ%20Protocol.pdf and by *Jolly et al. (2016)*. Briefly, coronal brain slices 20 µm thick were collected and incubated with digoxigenin (DIG)- and/or fluorescein (FITC)-labelled RNA probes. For single probe ISH, the DIG signal was visualized with alkaline phosphatase (AP)-conjugated anti-DIG Fab fragment and either a mixture of nitroblue tetrazolium (NBT, Roche) and 5-bromo-4-chloro-3-indolyl phosphate (toluidine salt) (BCIP, Roche) for chromogenic reaction or Fast Red fluorescence system (Roche). For double ISH, two probes – one FITC- and the other DIG-labelled – were applied simultaneously. For *Gria* (DIG)/ *Pdgfra* (FITC) probe combinations, the DIG-labelled probe was detected first using Fast Red, followed by detection of FITC-labelled probe with horseradish peroxidase (POD)-conjugated anti-FITC Fab Fragments (Roche) and tyramide signal amplification system (Perkin Elmer), according to the manufacturer's instructions. For *Gria* (DIG)/ *Enpp6* (FITC) probe combinations, DIG- and FITC-labelled probes were detected with POD-conjugated anti-DIG and anti-FITC Fab fragments, followed by tyramide signal amplification. FITC and Cyanine3 (Perkin Elmer) were used for anti-FITC-POD and anti-DIG-POD antibodies, respectively. After ISH, immunohistochemistry was performed as described below. The plasmids (IMAGE clones) used to generate RNA probes were: clone 6842391 (*Gria1*), 6494013 (*Gria2*), 30668476 (*Gria3*), 6409918 (*Gria4*) and 4237600 (*Enpp6*).

## Immunohistochemistry

Free-floating coronal sections (25 µm) were permeabilized/ blocked in a solution containing 10% (v/v) sheep, cow or donkey serum in 0.1% (v/v) Triton X-100 in PBS for 1–2 hr at 20–25°C. Sections were subsequently incubated in primary antibody diluted in blocking solution at 4°C overnight, washed in PBS and incubated with secondary antibody diluted in blocking solution containing Hoechst 33258 dye (Sigma, 1:1000) for 1–4 hr at 20–25°C. Primary antibodies were anti-Pdgfra (rabbit, New England Biolabs, 1:500 dilution), anti-GFP (chicken, Avas, 1:500 or rat, Fine Chemical Products Ltd., 1:3000), anti-APC (clone CC1; mouse, Calbiochem,1:200), anti-Olig2 (rabbit, Chemicon, 1:700), anti-human-Olig2 (goat, R and D, 1:100), anti-Sox10 (guinea pig, 1:2000; a gift from M. Wegner, University of Erlangen, Germany), anti-Ki67 (mouse, BD Biosciences, 1:1000), anti-NeuN (rabbit, Chemicon, 1:500) and anti-cleaved Caspase-3 (rabbit, Abcam, 1:200). Secondary antibodies (raised in donkey or goat) were Alexa Fluor 488-, 568-, 647-, or Cy3-conjugated secondary antibodies to rabbit, mouse, chicken, rat, human or guinea pig (Invitrogen,1:500–1000). EdU detection was performed immediately after immunohistochemistry using the AlexaFluor-555 Click-iT detection kit (Invitrogen).

## Image acquisition and analysis

Images were captured with either a light microscope (Zeiss Axioplan) with a Hamamatsu digital camera (10x or 20x objective) or with a laser scanning confocal microscope (Leica TCS SPE; 20x objective), using standard excitation and emission filters for Hoechst 33258 and Alexa fluorophores. Confocal Z-stacks were captured for each section with 1.5 µm steps.

Cells were counted in images of non-overlapping confocal sections of corpus callosum (3–4 fields covering the entire length between the dorsolateral corners of the lateral ventricles at bregma 0–1 mm) or motor cortex (3–4 fields covering all cortical layers in at least three sections per mouse; numbers of mice are indicated in Results). Images were displayed on-screen in Adobe Photoshop and cells were marked as they were counted, to avoid double-counting. All experiments were counted with the experimenter blinded as to genotype.

## Electron microscopy

Mice were perfused at UCL with PBS followed by 2.5% (v/v) glutaraldehyde and 2% (w/v) paraformaldehyde in 0.1 M cacodylate buffer (pH 7.4). Brains were removed into the same fixative, kept at 4°C for 48 hr, washed with 0.1 M cacodylate buffer then rinsed briefly with distilled water and shipped to Japan in PBS at ambient temperature. Sagitally-sectioned brains were processed for electron microscopy (EM). After immersion in a 1% (w/v) osmium tetroxide solution for 2 hr at 4°C, the specimens

were dehydrated through a graded alcohol series and embedded in Epon 812 (TAAB Laboratories, UK). The required area was trimmed and ultrathin sections were cut, collected on a platinum-coated glass slide, stained with uranyl acetate and lead citrate and imaged in a scanning electron microscope equipped with a back-scattered electron beam detector (Hitachi SU8010) at 1.5 kV accelerating voltage. EM images of sagittal sections of corpus callosum were taken at 2000x or 4000x magnification at P14, and 15000x at P70 starting 100 µm posterior to the anterior-most edge of the corpus callosum and spanning its entire dorso-ventral width, resulting in tiled images ~ 210 µm x 160 µm for P14 and ~25 µm x 170 µm for P70. Numbers of cross-sectional views of myelinated axons were counted in one such image per animal (~2000 axons counted in each image, n = 5 mice per genotype at P14 and >5000 axons counted in each image, n = 3 or four mice per genotype at P70). To estimate g-ratios, the circumference of the axon as well as the external circumference of the myelin sheath were measured using ImageJ, avoiding paranodal areas. The diameters of axon and myelin sheath were calculated from these circumferential measurements, assuming a circular profile, and the g-ratio was calculated. 50–100 axons were counted in each of 4 mice per genotype.

## Statistics

Prism 6.0 (GraphPad) was used for statistical analysis. Data are presented as mean ± s.e.m. The Shapiro-Wilk test was used to test for normality of data distribution and p-values were determined using a two-tailed Student's t-test (when the distribution was consistent with normality) or non-parametric Mann-Whitney U test, as appropriate. One-way ANOVA with Bonferroni post-hoc test or Kruskal-Wallis with Dunn post hoc test were used for comparing multiple groups, as appropriate. When multiple t-tests or Mann-Whitney tests were performed, p-values were corrected for multiple comparison using a procedure equivalent to the Holm–Bonferroni method (for $N$ comparisons, the most significant $P$ value is multiplied by $N$, the second most significant by $(N − 1)$, the third most significant by $(N − 2)$, etc.; corrected p-values<0.05 are considered significant). Numbers of animals and cells studied are stated for each experiment in the text.

We analyzed the Caspase-3 immunolabelling data as follows. First, we calculated for control conditions the average fraction of Olig2$^+$ cells that also expressed cleaved Caspase-3 (and hence were undergoing apoptosis), by averaging across control litters (these numbers are stated in the main text). Second, within each litter we normalized the data from experimental mice (double- or triple-knockouts) and their *Gria3$^{null}$* controls to the average of the controls in that same litter, setting the latter to 1. Then we averaged these normalized data across all litters to obtain the mean percentage increase in the fraction of Olig2$^+$ cells that were apoptotic in experimental mice compared to control mice.

We did not perform a priori power analyses to calculate the sample size for the experiments, because the size of the effects expected was unknown before doing the experiments. We did however perform sample size calculation post hoc for experiments where group differences were close to significant, using the observed mean ± s.d. of the control group, an effect size of 20–25%, a power of 80% and p-value<0.05. With n = 3 for both groups, we should be able to detect a difference of ~23% in CC1$^+$ OLs in *Gria2$^{-/-}$* (**Figure 4H**). With n = 9 for control and n = 7 for the knockout group we should be able to detect a ~20% reduction in CC1$^+$ OLs in P21 *Gria3$^{null}$2$^{-/-}$* (**Figure 7B**). With n = 14 for both groups, we should be able to detect a difference of ~20% in P14 Pdgfra$^+$ OPs in *Gria3$^{null}$2$^{-/-}$* (**Figure 7F**).

## Acknowledgements

We thank Matthew Grist and Ulla Dennehy for expert technical help, and Nathan Skene, Marcio Oliveira and Sarah Jolly for advice and support. We thank Katsutoshi Ogasawara and all the staff of the Center for Electron Microscopy and Bio-Imaging Research, Iwate Medical University, for technical help. Work in WDR's laboratory is funded by the European Research Council (grant agreement 293544) and the Wellcome Trust (100269/Z/12/Z and 108726/Z/15/Z). Work in DA's laboratory is supported by the Wellcome Trust (099222/Z/12/Z). Work in RS's laboratory is funded by the Max Planck Society and the German Research Foundation (SFB636/A4, SFB1134/B01 and SFB 1158/A05). KT was supported by the Japanese Ministry of Education, Culture, Sports, Science and Technology and the Japanese Society for the Promotion of Science (25245069 and 24650181). EK was supported by a studentship from the Wellcome Trust. *Sox10-Cre* mice are available from Jackson Laboratories

(stock 025807). *Gria2^flox* and *Gria3^null* mice are available from the European Mouse Mutant Archive (strains EM:09212 and EM:09215, respectively).

## Additional information

### Funding

| Funder | Grant reference number | Author |
| --- | --- | --- |
| Wellcome | 100269/Z/12/Z | William D Richardson |
| Wellcome | 108726/Z/15/Z | William D Richardson |
| Wellcome | 099222/Z/12/Z | David Attwell |
| Wellcome | 089591/Z/09/Z | Eleni Kougioumtzidou |
| European Research Council | 293544 | William D Richardson |
| Deutsche Forschungsgemeinschaft | SFB636/A4 | Rolf Sprengel |
| Deutsche Forschungsgemeinschaft | SFB1134/B01 | Rolf Sprengel |
| Deutsche Forschungsgemeinschaft | SFB 1158/A05 | Rolf Sprengel |
| Japan Society for the Promotion of Science | 25245069 | Koujiro Tohyama |
| Japan Society for the Promotion of Science | 24650181 | Koujiro Tohyama |

The funders had no role in study design, data collection and interpretation, or the decision to submit the work for publication.

### Author contributions

EK, Conceptualization, Data curation, Formal analysis, Investigation, Methodology, Writing—original draft, Project administration; TS, NBH, KT, Formal analysis, Investigation, Methodology; RS, HM, Resources; DA, Resources, Data curation, Supervision, Funding acquisition, Writing—review and editing; WDR, Conceptualization, Resources, Data curation, Supervision, Funding acquisition, Project administration, Writing—review and editing

### Author ORCIDs

David Attwell, http://orcid.org/0000-0003-3618-0843

William D Richardson, http://orcid.org/0000-0001-7261-2485

### Ethics

Animal experimentation: All animal experiments were pre-approved by the UCL Ethical Committee and authorized by the Home Office of the UK Government in accordance with Animals (Scientific Procedures) Act 1986., under Project Licences PPL 70/7299 and PPL 70/8976 (D. Attwell) and PPL 70/7614 (W.D. Richardson).

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
