## [Decision Letter]

Thank you for submitting your work entitled "AMPA-type glutamate receptors on oligodendrocyte precursors modulate development of oligodendrocytes and myelin" for consideration by *eLife*. Your article has been reviewed by three peer reviewers, and the evaluation has been overseen by a Reviewing Editor and a Senior Editor.

The reviewers appreciated that your exploration of the issue of how neuronal activity and glutamate signaling regulate development of the oligodendrocyte lineage is timely and important. Unfortunately, however, they found the interpretation of the effects of inactivation of AMPA receptors gluR2/3 on oligodendrocyte maturation, myelin/myelinated axons, and AMPA currents somewhat preliminary. The reviews are summed below, but the main, shared concerns include:

1) Whether the transient decrease in CC1+ oligodendrocytes arises due to a delay in differentiation or a decrease in survival.

2) Whether myelination is truly decreased/delayed; suggestions for further analyses are to examine parameters of myelin internodes and to better quantify the EM data.

3) Whether in the absence of GluA2, alterations in AMPA current are due to AMPAR inhibition or to altered AMPAR signaling. In line with this, reviewer 2 had concerns that AMPAR-mediated current still remains in the double knockouts, potentially obscuring some effects of AMPAR inhibition, and thus producing only a modest change.

Because your study requires additional amendments that would likely take longer than a few months, we will have to reject the manuscript at this time.

*Reviewer #1:*

In the manuscript by Kougioumtzidou et al., the authors set out to investigate whether AMPAR signaling has a role in regulating oligodendrocyte development in vivo (first demonstration). They show that oligodendrocyte precursor cells express 3 of 4 known AMPAR subunits (Gria2-4). They then proceed to analyze oligodendrocyte development in Gria3 KOs, SOX10-conditional Gria2 KOs and Gria2/3 double KOs. They show that there are no effects on oligodendrocyte development in the individual KOs, even though AMPAR signaling is affected. However, the double KOs show an early, but transient decrease/delay in differentiated oligodendrocyte density during development (P14), which returns to normal by P21, and they conclude that AMPAR signaling regulates oligodendrocyte differentiation. The mice display a similar decrease in the number of myelinated axons in the double KOs and conclude that AMPAR signaling also regulates myelination. Since glutamatergic signaling through AMPAR is thought to be a major form of communication between neurons and oligodendrocyte precursors, and the resulting effects of this communication on oligodendrocyte development are still largely unknown, the topic of this manuscript is timely and highly important to the field. However, the work presented falls far short of accomplishing its aim and does little to move the field forward. Major concerns are outlined below.

1) The authors illustrate that a good deal of AMPAR-mediated current (~50%) still remains in the double KOs, which may obscure some effects of AMPAR inhibition. Perhaps because of this the effects demonstrated in this manuscript as a result of the double KOs are modest and do not add any mechanistic insight to what is already known.

2) As the authors state, the fact that there is no change in OP densities/proliferation in Gria2/3 KOs argues against the decrease in CC1+ OLs being due to a delay in developmental OL differentiation, but more for a decrease in OL survival, which the authors were unable to quantify. Not only does this leave the work incomplete, but it also makes it improper to conclude that AMPARs "modulate the development of OLs" when the effects seen could be a survival issue.

3) The authors show a reduction in myelinated neurons and suggest that OL differentiation and therefore myelination is delayed. However, they also present g-ratios that are unaffected. If developmental myelination is indeed delayed the KOs would show increased g-ratios since the myelin would begin to form later than controls. No change in g-ratios suggests other possibilities such as a survival issue of the OLs (see #2), or even the possibility that myelin internode lengths are shorter. Other studies have shown that OLs make shorter myelin sheaths upon inhibition of neuronal activity and decreased glutamate signaling (Hines et al. 2015. Nat Neurosci. 18:683-689; Mensch et al. 2015. Nat Neurosci. 18:628-630., Etxeberria et al. 2016. J Neurosci. 36:6037-6948.), therefore, it is a likely possibility that the decrease in myelinated axons observed by the authors is due to a similar change in myelin dynamics. An in-depth analysis of the myelin internodes (length and number/OL) is required to ascertain the full effect of AMPR function. Further investigation is necessary.

4) It is known that the GluA2 subunit makes AMPARs impermeable to Ca+2, therefore knocking out Gria2 will lead to increased Ca+2 signaling and changes the whole nature of AMPAR signaling. Moreover, the authors suspect that the remaining AMPAR-mediated currents are due to homotetramers of GluA4, but these are likely never normally produced in OPCs and may not reveal any meaningful signaling effects. Ultimately it is not clear whether the effects the authors are studying are due to AMPAR inhibition or to altered AMPAR signaling.

*Reviewer #2:*

Oligodendrocyte lineage cells express glutamate receptors. Nevertheless, the functional significance of the expression of the distinct types of glutamate receptors by these cells has not been completely characterized. In the current study Kougioumtzidou and colleagues sought to identify the role that AMPA receptors play in oligodendrocyte lineage cells. They found that kainate-induced current is mediated by the expression of the AMPA receptors GluA2, GluA3 and GluA4 in oligodendrocyte lineage cells. In addition, these investigators genetically inactivated GluA2 and GluA3, individually and together, to determine the functional significance of this expression. The investigators took advantage of pre-existing GluA3 null mutant mice, as well as a GluA2 conditional mutant mouse line that they combined with a Sox10-Cre driver to promote recombination in oligodendrocyte lineage cells. This resulted in mice in which GluA3 expression was omitted from all cells, whereas GluA2 was specifically eliminated from oligodendrocyte lineage cells, and the expression of GluA4 remained intact. Although the individual mutants were unaffected in a detectable way, the double mutant mice exhibited partial blockage of kainate-induced current in oligodendrocyte progenitor cells. In agreement with previous studies, the authors found that the partial blockage of the kainate- induced current in oligodendrocyte progenitor cells did not affect these cells in any identifiable manner. Interestingly, however, the authors demonstrate that the combined ablation of GluA2 and GluA3 resulted in an approximate twenty percent reduction in the number of mature oligodendrocytes at early postnatal ages that correlates with mild hypomyelination. Overall, this is an interesting and well-done study that adds to our growing understanding of the role that glutamate signaling plays in oligodendrocyte lineage cells.

1) Oligodendrocyte progenitor cells are highly proliferative. Since the authors use a conditional approach to the inactivation of GluA2, it is likely that not all OPCs have both alleles of GluA2 inactivated. It is possible that over time OPCs with the conditional inactivation of GluA2 are selected against, which might explain the observed recovery in older animals. A more detailed examination of this would be very informative: is there a higher percentage of WT GluA2 OPCs in older double-mutant animals?

2) A very interesting finding of this report is that AMPA receptors are important for differentiated, myelinating oligodendrocytes. Nevertheless, in my view this finding is not highlighted sufficiently. A proposed role for GluA2 and GluA3 in myelinating oligodendrocytes should be discussed.

3) Does the percent of myelinated axons recover to normal levels in adult animals? This data should be included in Figure 8.

*Reviewer #3:*

Kougioumtzidou et al., submit "AMPA-type glutamate receptors on oligodendrocyte precursors modulate development of oligodendrocytes and myelin" for consideration at *eLife*. This work centres on to topic of how neuronal activity regulates development of the oligodendrocyte lineage. This general area will be of broad interest to the readership of *eLife* in large part because changes to oligodendrocyte development and myelination help regulate or fine tune neural circuit function and behaviour – the latter shown in a very elegant study by the Richardson lab. In this study, the authors address how glutamate signalling directly impacts oligodendrocyte development in vivo (a long-standing question), with a focus on AMPAR mediated function. Although one might argue that the authors did not address how AMPAR function in its entirety affects oligodendrocyte development (disrupting gluR2 and gluR3, but not gluR4, which is also expressed in the OL lineage), they do see a clear phenotype in vivo, and this is of interest in and of itself. However, at the moment I think that the manuscript would benefit greatly from some additional work to allay a few concerns, and to strengthen some of the conclusions drawn.

Validation of gria3 global mutant as reasonable "control" background. The authors go to some lengths to highlight the importance of a conditional targeting approach, whereby AMPRA mediated signalling/ function is disrupted specifically in oligodendrocytes. They achieve this is by targeting gluR2 function using floxed alleles of gria2 and a sox10-cre, which seems sufficiently faithful to the oligodendrocyte lineage in the CNS. However, all experiments are done on a background of lack of GluR3 function, i.e. a germline gria3 mutant mouse. Indeed all "controls" are gria3 mutant. The authors do show that gria3 mutants exhibit no change in PDGFRa expressing oligodendrocyte precursors (or their proliferation) or CC1 expressing mature oligodendrocytes, but only one stage is examined, and no further tests are carried out. In addition to the possibility of gria3 mutant oligodendrocytes having defects at other stages or at other times, these mutants will surely have compensation for glur3 loss by expression/ regulation of other GluR subunit isoforms and disruption to neural circuit function that could potentially confound analysis of myelination. It would be reassuring to see further analyses of gria3 mutants compared to controls, e.g. additional stages, electron microscopy (see last point below), to allay concerns that its use as a control is not confounding.

Mechanism by which reduction in oligodendrocyte number is manifest

It is certainly interesting that there is a reduction (albeit transient) in mature oligodendrocyte number and enpp6 expressing cell number, but the follow-up analyses as to how this comes to be come across as being a little preliminary. The authors report no significant change in PDGFRa density, but there is a stage at which there is a trend reduction in double mutants with a p value of 0.06. Could an increase in "n" unmask a phenotype? The analyses of cell death are also unsatisfying, given the importance of clarifying the role for AMPAR in the cell lineage. The use of additional marker(s) for cell death would be important. Is there no constitutive oligodendrocyte lineage cell death in actual wildtype animals at any of the stages examined?

Fewer synapses exist in double mutants. The conclusion that oligodendrocyte precursors have fewer synapses per cell is an inference based on the physiological analyses. Although a reasonable inference, to provide independent evidence in support of this conclusion would be good. Could the authors try to visualise synapse number in OPs, e.g. with a presynaptic marker at least, or ideally with a pre and post-synaptic marker together with NG2, enpp6 etc.?

Electron microscopy analyses could be better quantified. It is important that the electron microscopy data support the observations of a decrease in oligodendrocyte number, whereby the authors observe a corresponding decrease in relative number of axons that are myelinated. It would be helpful to assess axonal sizes and distributions in the mutants as well, and ideally compare with controls that were not gria3 null. Comparison of gria3 mutants to control littermates would suffice.

[Editors’ note: what now follows is the decision letter after the authors submitted for further consideration.]

Thank you for resubmitting your work entitled "Signalling through AMPA receptors on oligodendrocyte precursors promotes oligodendrocyte survival and myelination" for further consideration at *eLife*. Your revised article has been favorably evaluated by Marianne Bronner (Senior Editor), a Reviewing Editor and three reviewers.

The three reviewers agreed that the manuscript is greatly improved. They applaud your success in making the AMPAR triple KO's that have an abrogation of almost all AMPAR-related currents in the oligodendrocyte lineage. They consider your finding that the disruption to AMPAR signaling in oligodendrocytes causing their increased cell death is sufficient to explain the subsequent deficits in myelination, a strong and exciting addition to the field.

There are a few remaining issues that need to be addressed before acceptance, mostly textual, as outlined below:

1) Title: From your data, it seems as though the role of AMPAR signaling in OPCs/ immature ENPP6 expressing OLs is to control the number of oligodendrocytes that can be maintained and employed for myelination or discarded, if unnecessary. The title stating "survival and myelination" implies two separate roles; "survival and thus myelination" would be a more apt reference to your findings.

2) Data on the cell death phenotype (Figure 7 and Figure 10): To support this finding, it would be helpful to understand the proportion of cells that are caspase positive in the population examined in your analyses, to understand what level of possible overproduction/regulation by activity is at play in the system. Even though the numbers are likely not absolute, if you thought it reasonable, adding these numbers would further bolster your case.

3) Discussion:

A) The lack of an obvious role for AMPAR in regulating myelin sheath number, length, thickness per cell is of great interest. However, it may be misleading to state that this result mirrors observations that in vitro sheath length is not regulated by activity. Indeed, evidence from the Fields lab shows that inactive axon sheath lengths are significantly shorter on silent (vesicular release impaired) axons (Wake et al., Nat Comm), and similarly, in vivo in zebrafish (Koudelka et al., Curr Biol.) The AMPAR negative data may instead imply that control of sheath length is mediated by other glutamatergic receptors, or by other signals, and should be noted.

B) It may be inaccurate to state that reduction of activity did not affect OL numbers in zebrafish with reduced neuronal activity; a modest reduction in cell number was seen with reduced activity, and conversely an increase with increase in activity (Mensch et al., Nat Neurosci). It would be helpful to address control of cell number at specific stages of OL development, given the extensive work on this topic and recent work identifying other possible regulators of OPC proliferation (Venkatesh et al., Cell). Discuss whether the proportion regulated by AMPAR reflects cells that have transitioned from pure OPCs to ENPP6-expressing stage and are faced with an activity-regulated decision to commit to myelination, thus a time-specific role for AMPAR in the OL lineage.

---

## [Author Response]

[Editors’ note: the author responses to the first round of peer review follow.]

[…] Reviewer #1:

*In the manuscript by Kougioumtzidou et al., the authors set out to investigate whether AMPAR signaling has a role in regulating oligodendrocyte development in vivo (first demonstration). They show that oligodendrocyte precursor cells express 3 of 4 known AMPAR subunits (Gria2-4). They then proceed to analyze oligodendrocyte development in Gria3 KOs, SOX10-conditional Gria2 KOs and Gria2/3 double KOs. They show that there are no effects on oligodendrocyte development in the individual KOs, even though AMPAR signaling is affected. However, the double KOs show an early, but transient decrease/delay in differentiated oligodendrocyte density during development (P14), which returns to normal by P21, and they conclude that AMPAR signaling regulates oligodendrocyte differentiation. The mice display a similar decrease in the number of myelinated axons in the double KOs and conclude that AMPAR signaling also regulates myelination. Since glutamatergic signaling through AMPAR is thought to be a major form of communication between neurons and oligodendrocyte precursors, and the resulting effects of this communication on oligodendrocyte development are still largely unknown, the topic of this manuscript is timely and highly important to the field. However, the work presented falls far short of accomplishing its aim and does little to move the field forward.*

We agree that the topic is important and timely and we trust that the large amount of extra work we have done on the project since its first submission, including the addition of *Gria2/3/4* triple-‐ knockouts, will convince this reviewer that our study is now a convincing and useful contribution to the field. Major additions to the manuscript since the first submission are listed above and include experiments demonstrating that:

i) AMPAR-‐mediated currents and EPSCs are almost abolished when *Gria2/3/4* are knocked out;

ii) by increasing OL death, this reduces the number of OLs by ~25%.

Thus, the effect of AMPAR activation in OL lineage cells is precisely the opposite of what was expected from cell culture work by the Gallo group 2 decades ago, which showed that AMPAR/KAR activation stopped OP proliferation. Therefore our paper significantly alters our understanding of how the development of the OL lineage is regulated by glutamatergic signalling.

*Major concerns are outlined below.*

*1) The authors illustrate that a good deal of AMPAR-mediated current (~50%) still remains in the double KOs, which may obscure some effects of AMPAR inhibition. Perhaps because of this the effects demonstrated in this manuscript as a result of the double KOs are modest and do not add any mechanistic insight to what is already known.*

We have now knocked out *Gria4* conditionally in the OL lineage in addition to *Gria2* and *Gria3*, reducing the AMPAR-mediated current to ~10% of control and the frequency of EPSCs to practically zero. This produces a ~25-30% reduction in the number of OLs and this reduction is no longer transient as in the double-KO but persists into young adulthood (at least P53). These new data are shown in two new Figure 9 and Figure 10, and described in the subsection “Triple knockout of GluA2/ GluA3/ GluA4 in the OL lineage”. In addition we now show (by Caspase-3 immunolabelling) that this phenotype is caused by an increase in the death of newly-forming OLs (see our response to this reviewer’s point 2, below). Thus, our study advances the field considerably by providing the first convincing demonstration of the role of AMPARs in regulating OL number in vivo.

*2) As the authors state, the fact that there is no change in OP densities/proliferation in Gria2/3 KOs argues against the decrease in CC1+ OLs being due to a delay in developmental OL differentiation, but more for a decrease in OL survival, which the authors were unable to quantify. Not only does this leave the work incomplete, but it also makes it improper to conclude that AMPARs "modulate the development of OLs" when the effects seen could be a survival issue.*

We have now used Caspase-3 immunolabelling to demonstrate an increase in the rate of developmental death among newly-forming OLs in the AMPAR knockouts. These new data are depicted for the *Gria2/3* double-KO in Figure 7 and described in the fourth paragraph of the subsection “OL survival is reduced, and OL accumulation delayed, in *Gria3^null^2^-/-^* white matter”. For the *Gria2/3/4* triple‐KO the data are shown in Figure 10 and described in the last paragraph of the subsection “Triple knockout of GluA2/ GluA3/ GluA4 in the OL lineage”.

*3) The authors show a reduction in myelinated neurons and suggest that OL differentiation and therefore myelination is delayed. However, they also present g-ratios that are unaffected. If developmental myelination is indeed delayed the KOs would show increased g-ratios since the myelin would begin to form later than controls. No change in g-ratios suggests other possibilities such as a survival issue of the OLs (see #2), or even the possibility that myelin internode lengths are shorter. Other studies have shown that OLs make shorter myelin sheaths upon inhibition of neuronal activity and decreased glutamate signaling (Hines et al. 2015. Nat Neurosci. 18:683-689; Mensch et al. 2015. Nat Neurosci. 18:628-630., Etxeberria et al. 2016. J Neurosci. 36:6037-6948.), therefore, it is a likely possibility that the decrease in myelinated axons observed by the authors is due to a similar change in myelin dynamics. An in-depth analysis of the myelin internodes (length and number/OL) is required to ascertain the full effect of AMPR function. Further investigation is necessary.*

We took the reviewer’s comments to heart and analyzed the morphology of individual OLs by dyefilling cells in brain slices. We found that there is no detectable difference in either internode number per OL or internode length in our KO mice. These data are shown in Figure 8 for the double-KO and Figure 10 for the triple-KO, and described in the first paragraph of the subsection “Reduced total myelin in *Gria3^null^2^-/-^* white matter” for the double‐KO and in the last paragraph of the subsection “Triple knockout of GluA2/ GluA3/ GluA4 in the OL lineage” for the triple-KO. It is therefore likely that, once a commitment is made to myelinate an axon, the process continues until a normal g ratio is attained. This has now been stated in the fourth paragraph of the subsection “AMPAR signalling promotes the survival of differentiating OLs”.

*4) It is known that the GluA2 subunit makes AMPARs impermeable to Ca+2, therefore knocking out Gria2 will lead to increased Ca+2 signaling and changes the whole nature of AMPAR signaling. Moreover, the authors suspect that the remaining AMPAR-mediated currents are due to homotetramers of GluA4, but these are likely never normally produced in OPCs and may not reveal any meaningful signaling effects. Ultimately it is not clear whether the effects the authors are studying are due to AMPAR inhibition or to altered AMPAR signaling.*

We understand how the Ca^2+^ permeability of AMPARs is altered in the absence of GluA2, and indeed used this property as part of our electrophysiological characterization of the *Gria2* knockout.

However, the fact that we did not observe any phenotype of *Gria2^–/–^* OPs relative to *Gria2^+/–^* or *Gria2^+/+^* littermates (Figure 4) rules out a specific role for AMPAR-mediated Ca^2+^ signalling. Rather, the reduction in OL number observed in *Gria2/3* and *Gria2/3/4* mutants would appear to result from another aspect of AMPAR signaling, for example the depolarization or Na^+^ influx that it produces. This is now discussed in the first paragraph of the Discussion. We agree with the reviewer that GluA4 homo‐tetramers might not normally form in wild type OPs, but they do appear to form in our *Gria2/3* double‐KOs, as shown by the fact that the residual kainate‐induced AMPAR current in the double-KO is obliterated when GluA4 is also removed. The more severe phenotype of the triple-KO compared to the double-KO (i.e. persistence of the phenotype into adulthood) and, by extension, the phenotype of the *Gria2/3* KOs themselves, thus appears to be caused by an inhibition of AMPAR signaling other than an alteration in Ca^2+^ permeability.

Reviewer #2:

*[…] 1) Oligodendrocyte progenitor cells are highly proliferative. Since the authors use a conditional approach to the inactivation of GluA2, it is likely that not all OPCs have both alleles of GluA2 inactivated. It is possible that over time OPCs with the conditional inactivation of GluA2 are selected against, which might explain the observed recovery in older animals. A more detailed examination of this would be very informative: is there a higher percentage of WT GluA2 OPCs in older double-mutant animals?*

This is an interesting point. We have not managed to identify anti-GluA2 antibodies that would allow us to quantify the fraction of Pdgfra^+^ OPCs that are GluA2^+^ as a function of age in the double‐KOs. However, we think this question has been overtaken by our generating the *Gria2/3/4* triple-KO, the phenotype of which extends at least into early adulthood (P53^+^). This is shown in Figure 10.

*2) A very interesting finding of this report is that AMPA receptors are important for differentiated, myelinating oligodendrocytes. Nevertheless, in my view this finding is not highlighted sufficiently. A proposed role for GluA2 and GluA3 in myelinating oligodendrocytes should be discussed.*

We have shown that the number of differentiated OLs that survive and form myelin is reduced in the *Gria2/3* double-KO and *Gria2/3/4* triple‐KO. As Dr Popko points out, this suggests that AMPAR signalling is important in early‐differentiating OLs, implying that GluA subunits should be expressed in pre‐myelinating OLs (*Enpp6^+^*) as well as in their precursors. We therefore performed double-fluorescence ISH for *Gria24* and *Enpp6* and found that there was substantial overlap (Figure 7—figure supplement 1, described in the last paragraph of the subsection “OL survival is reduced, and OL accumulation delayed, in *Gria3^null^2^–/–^* white matter”). As requested, we have now discussed the role of AMPARs in early myelinating OLs in the third paragraph of the subsection “AMPAR signalling promotes the survival of differentiating OLs”.

*3) Does the percent of myelinated axons recover to normal levels in adult animals? This data should be included in Figure 8.*

We have extended our EM analysis, including an investigation of this point, and now show that the density of myelin figures does recover to normal levels by P70 in *Gria2/3* double-KOs. This is shown in Figure 8 and described in the last paragraph of the subsection “Reduced total myelin in *Gria3^null^2^–/–^* white matter”.

Reviewer #3:

*Kougioumtzidou et al., submit "AMPA-type glutamate receptors on oligodendrocyte precursors modulate development of oligodendrocytes and myelin" for consideration at eLife. This work centres on to topic of how neuronal activity regulates development of the oligodendrocyte lineage. This general area will be of broad interest to the readership of eLife in large part because changes to oligodendrocyte development and myelination help regulate or fine tune neural circuit function and behaviour – the latter shown in a very elegant study by the Richardson lab. In this study, the authors address how glutamate signalling directly impacts oligodendrocyte development in vivo (a long-standing question), with a focus on AMPAR mediated function. Although one might argue that the authors did not address how AMPAR function in its entirety affects oligodendrocyte development (disrupting gluR2 and gluR3, but not gluR4, which is also expressed in the OL lineage), they do see a clear phenotype in vivo, and this is of interest in and of itself. However, at the moment I think that the manuscript would benefit greatly from some additional work to allay a few concerns, and to strengthen some of the conclusions drawn.*

In retrospect, we accept the reviewer’s sense that the initial submission was a little preliminary and we have worked hard in the meantime to address his/her concerns and those of the other reviewers and to strengthen our conclusions.

*Validation of gria3 global mutant as reasonable "control" background. The authors go to some lengths to highlight the importance of a conditional targeting approach, whereby AMPRA mediated signalling/ function is disrupted specifically in oligodendrocytes. They achieve this is by targeting gluR2 function using floxed alleles of gria2 and a sox10-cre, which seems sufficiently faithful to the oligodendrocyte lineage in the CNS. However, all experiments are done on a background of lack of GluR3 function, i.e. a germline gria3 mutant mouse. Indeed all "controls" are gria3 mutant. The authors do show that gria3 mutants exhibit no change in PDGFRa expressing oligodendrocyte precursors (or their proliferation) or CC1 expressing mature oligodendrocytes, but only one stage is examined, and no further tests are carried out. In addition to the possibility of gria3 mutant oligodendrocytes having defects at other stages or at other times, these mutants will surely have compensation for glur3 loss by expression/ regulation of other GluR subunit isoforms and disruption to neural circuit function that could potentially confound analysis of myelination. It would be reassuring to see further analyses of gria3 mutants compared to controls, e.g. additional stages, electron microscopy (see last point below), to allay concerns that its use as a control is not confounding.*

We have done a more searching analysis of the *Gria3* null mice, comparing an additional stage (P21) of *Gria3* null mice to WT controls and performing EM analysis of these mice at P14. We found no difference between *Gria3^null^* and wild type mice in these regards, reinforcing our conclusion that *Gria3^null^* mice are suitable controls. This is shown in Figure 3, and described in the first paragraph of the subsection “GluA2 and GluA3 are individually dispensable for OL production”.

*Mechanism by which reduction in oligodendrocyte number is manifest*

*It is certainly interesting that there is a reduction (albeit transient) in mature oligodendrocyte number and enpp6 expressing cell number, but the follow-up analyses as to how this comes to be come across as being a little preliminary. The authors report no significant change in PDGFRa density, but there is a stage at which there is a trend reduction in double mutants with a p value of 0.06. Could an increase in "n" unmask a phenotype? The analyses of cell death are also unsatisfying, given the importance of clarifying the role for AMPAR in the cell lineage. The use of additional marker(s) for cell death would be important. Is there no constitutive oligodendrocyte lineage cell death in actual wildtype animals at any of the stages examined?*

We have addressed these points in two ways. First, we have added 3 more animals to each group at the P14 stage and the Pdgfra^+^ precursor cell density still did not show a significant difference (p=0.07, described in the third paragraph of the subsection “OL survival is reduced, and OL accumulation delayed, in *Gria3^null^2^–/–^*white matter” and shown in Figure 7). Second, we have now generated *Gria2/3/4* triple‐KOs and found no detectable effect on Pdgfra^+^ precursor cell number at P14 or P53 (Figure 10), but a significant reduction in the number of mature (CC1+) OLs both at P14 and P53 (Figure 10), and described in the last paragraph. Thus, the lack of an effect on OP number is reproduced in the triple‐KO, but the OL phenotype of the triple-KO is more severe than the double-KO; it is no longer transient but extends into (at least) young adulthood. To get at the underlying mechanism we have analyzed apoptotic cell death in the OL lineage by Caspase-3/ Olig2 immunolabeling and found a significant increase in double- and triple-KOs relative to controls (Figure 7 and Figure 10). Therefore we can now say that the decreased OL cell number in the KOs is due to increased death of newly-forming OLs. In answer to the reviewer’s question about cell death in wild type animals – yes there is significant death in the controls as expected but this is increased by ~20% in the KOs, as shown in Figure 7 and Figure 10.

*Fewer synapses exist in double mutants. The conclusion that oligodendrocyte precursors have fewer synapses per cell is an inference based on the physiological analyses. Although a reasonable inference, to provide independent evidence in support of this conclusion would be good. Could the authors try to visualise synapse number in OPs, e.g. with a presynaptic marker at least, or ideally with a pre and post-synaptic marker together with NG2, enpp6 etc.?*

We absolutely agree that it is important to strive to visualize axon-OP synapses in vivo. We had previously (before the original submission) tried hard to visualize synapses in the subcortical white matter by immunolabelling for PSD95 and VGlut1, but without success immunolabelling in white matter is difficult at the best of times and often the individual requirements of different antibodies cannot be reconciled. Furthermore, we cannot be sure whether the removal of synaptic input seen in the KOs is due to a physical loss of the synapse, or a simple absence of postsynaptic AMPARs so that the synapse becomes “silent”. Thus, we feel strongly that our electrophysiology data provide sufficient information for the present purpose and that anything further is beyond the scope of this study. In the triple-KOs we find that the frequency of Ruthenium Red‐stimulated EPSCs is reduced to zero, along with a ~90% reduction in total kainate-induced current, implying that there are no longer any functional AMPAR-containing synapses in OPs, as expected. It seems very likely that the OL phenotypes we describe result from reduction and loss of these synapses in the double- and triple-KOs, respectively. This has now been stated in the first paragraph of the subsection “AMPAR signalling promotes the survival of differentiating OLs”

*Electron microscopy analyses could be better quantified. It is important that the electron microscopy data support the observations of a decrease in oligodendrocyte number, whereby the authors observe a corresponding decrease in relative number of axons that are myelinated. It would be helpful to assess axonal sizes and distributions in the mutants as well, and ideally compare with controls that were not gria3 null. Comparison of gria3 mutants to control littermates would suffice.*

As suggested by the reviewer, we have now measured the diameters of myelinated and unmyelinated axons in the corpus callosum of *Gria3* null mice versus wild type controls. We found no differences in the range or frequency distribution of axon diameters, or in the g-ratio shown in Figure 3‐J and described in the first paragraph of the subsection “GluA2 and GluA3 are individually dispensable for OL production”. Together with the analyses of the *Gria3* OLs described above in our response to this reviewer’s first major point, these data now provide strong evidence that loss of GluA3 in neurons and OPs does not affect development of the OL lineage. We also compared axon diameters and g-ratios in *Gria3^null^Gria2^–/–^* corpus callosum versus *Gria3^null^Gria2^+/+^* littermate controls, and again found no differences shown in Figure 8 and described in the first paragraph of the subsection “Reduced total myelin in *Gria3^null^2^–/–^* white matter”.

[Editors' note: the author responses to the re-review follow.]

*There are a few remaining issues that need to be addressed before acceptance, mostly textual, as outlined below:*

*1) Title: From your data, it seems as though the role of AMPAR signaling in OPCs/ immature ENPP6 expressing OLs is to control the number of oligodendrocytes that can be maintained and employed for myelination or discarded, if unnecessary. The title stating "survival and myelination" implies two separate roles; "survival and thus myelination" would be a more apt reference to your findings.*

We have changed the title to avoid giving the impression that AMPAR signalling has separate effects on oligodendrocyte survival and myelination. Our suggested new title is "Signalling through AMPA receptors on oligodendrocyte precursors promotes myelination by enhancing oligodendrocyte survival".

*2) Data on the cell death phenotype (Figure 7 and Figure 10): To support this finding, it would be helpful to understand the proportion of cells that are caspase positive in the population examined in your analyses, to understand what level of possible overproduction/regulation by activity is at play in the system. Even though the numbers are likely not absolute, if you thought it reasonable, adding these numbers would further bolster your case.*

We now state the average fraction of Olig2^+^ cells that express cleaved Caspase-3 in control conditions in the relevant sections of Results (subsection “OL survival is reduced, and OL accumulation delayed, in *Gria3^null^2^–/–^* white matter”, last paragraph and subsection “Triple knockout of GluA2/ GluA3/ GluA4 in the OL lineage”, last paragraph). This fraction was ~23% and was increased by a factor of ~1.19 (i.e. increased by ~19%) in the double-KO and by a factor of ~1.24 (increased by ~24%) in the triple KO. We have added a short paragraph to the Methods section (subsection “Statistics”, second paragraph) to explain how we analyzed these data.

*3) Discussion:*

*A) The lack of an obvious role for AMPAR in regulating myelin sheath number, length, thickness per cell is of great interest. However, it may be misleading to state that this result mirrors observations that in vitro sheath length is not regulated by activity. Indeed, evidence from the Fields lab shows that inactive axon sheath lengths are significantly shorter on silent (vesicular release impaired) axons (Wake et al., Nat Comm), and similarly, in vivo in zebrafish (Koudelka et al., Curr Biol.) The AMPAR negative data may instead imply that control of sheath length is mediated by other glutamatergic receptors, or by other signals, and should be noted.*

We have re-emphasized the findings of others that neuronal activity can regulate internode length, and cited the papers mentioned by the referee, as follows: "Therefore, the activity-regulation of internode length that has been observed in vitro (Wake et al., 2015) and in vivo (Koudelka et al., 2016) is presumably mediated by glutamate receptors other than AMPAR, or by signals other than glutamate."

*B) It may be inaccurate to state that reduction of activity did not affect OL numbers in zebrafish with reduced neuronal activity; a modest reduction in cell number was seen with reduced activity, and conversely an increase with increase in activity (Mensch et al., Nat Neurosci). It would be helpful to address control of cell number at specific stages of OL development, given the extensive work on this topic and recent work identifying other possible regulators of OPC proliferation (Venkatesh et al., Cell). Discuss whether the proportion regulated by AMPAR reflects cells that have transitioned from pure OPCs to ENPP6-expressing stage and are faced with an activity-regulated decision to commit to myelination, thus a time-specific role for AMPAR in the OL lineage.*

We have extended our discussion of the finding (Mensch et al., 2015) that OLs and OPs were reduced in number by ~10% in zebrafish following blockade of vesicular release by tetanus toxin, as follows: "Genetic block of vesicular release from longitudinally-projecting axons in the zebrafish spinal cord also had no effect on OP proliferation (Hines el al., 2015; Mensch et al., 2015), although numbers of OL lineage cells were reduced by ~10%, apparently through reduced specification of OPs from their stem cell precursors in the embryonic ventricular zone (Mensch et al., 201). It is unlikely that we could have detected an effect of this size, even assuming that it would not have been overridden by homeostatic controls on cell number in the perinatal mouse brain."